# Analyzing the turbulent Planetary Boundary Layer by remote sensing systems: Doppler wind lidar, aerosol elastic lidar and microwave radiometer

Gregori de Arruda Moreira[1,2,3], Juan Luis Guerrero-Rascado[1,2], Jose A. Benavent-Oltra[1,2], Pablo Ortiz-Amezcua[1,2], Roberto Román[1,2,4], Andrés E. Bedoya-Velásquez[1,2,5] Juan Antonio Bravo-Aranda[1,2], Francisco Jose Olmo Reyes[1,2], Eduardo Landulfo[3], Lucas Alados-Arboledas[1,2]

[1]Andalusian Institute for Earth System Research (IISTA-CEAMA), Granada, Spain
[2]Dpt. Applied Physics, University of Granada, Granada, Spain
[3]Institute of Research and Nuclear Energy (IPEN), São Paulo, Brazil
[4]Grupo de Óptica Atmosférica (GOA), Universidad de Valladolid, Valladolid, Spain.
[5]Sciences Faculty, Department of Physics, Universidad Nacional de Colombia, Medellín, Colombia.

*Correspondence to*: Gregori de Arruda Moreira (gregori.moreira@usp.br)

**Abstract**

The Planetary Boundary Layer ($PBL$) is the lowermost region of troposphere and endowed with turbulent characteristics, which can have mechanical and/or thermodynamic origins. Such behavior gives to this layer great importance, mainly in studies about pollutant dispersion and weather forecasting. However, the instruments usually applied in studies about turbulence in the $PBL$ have limitations in spatial resolution (anemometer towers) or temporal resolution (instrumentation onboard aircraft). Ground-based remote sensing, both active and passive, offers an alternative for studying the $PBL$. In this study we show the capabilities of combining different remote sensing systems (microwave radiometer [$MWR$], Doppler lidar [$DL$] and elastic lidar [$EL$]) for retrieving a detailed picture on the $PBL$ turbulent features. The statistical moments of the high frequency distributions of the vertical wind velocity, derived from $DL$ and of the backscattered coefficient derived from $EL$, are corrected by two methodologies, namely first lag and -2/3 correction. The corrected profiles, obtained from $DL$ data, present small differences when compared against the uncorrected profiles, showing the low influence of noise and the viability of the proposed methodology. Concerning $EL$, in addition to analyze the influence of noise, we explore the use of different wavelengths that usually include $EL$ systems operated in extended networks, like EARLINET, LALINET, MPLNET or SKYNET. In this way we want to show the feasibility of extending the capability of existing monitoring networks without strong investments or changes in their measurements protocols. Two case studies were analyzed in detail, one corresponding to a well-defined $PBL$ and another one corresponding to a situation with presence of a Saharan dust lofted aerosol layer and clouds. In both cases we discuss results provided by the different instruments showing their complementarity and the cautions to be applied in the data interpretation. Our study shows that the use of $EL$ at 532nm requires a careful correction of the signal using the first lag time correction in order to get reliable turbulence information on the $PBL$.

**Keywords:** Turbulence, Planetary Boundary Layer, Doppler lidar, elastic lidar, microwave radiometer, Earlinet.

## 1 Introduction

The Planetary Boundary Layer ($PBL$) is the atmospheric layer directly influenced by the Earth's surface that responds to its changes within time scales around an hour (Stull, 1988). Such layer is located at the lowermost region of troposphere, and is mainly characterized by turbulent processes and a daily evolution cycle. In an ideal situation, some instants after sunrise, the ground surface temperature increases due to the positive net radiative flux ($R_n$). This process intensifies the convection, where there is an ascension of warm air masses, causing the downward displacement of colder air masses and consequently originating the Convective Boundary Layer (CBL) or Mixing Layer (ML). Such layer has this name due to the mixing process generated by the ascending air parcels. Slightly before sunset, the gradual reduction of incoming solar irradiance at the Earth's surface causes the decrease of the positive $R_n$ and, consequently, its sign change. In this situation, there is a reduction of the convective processes and a weakening of the turbulence. In this process the $CBL$ leads to the development of two layers, namely a stably stratified boundary layer called Stable Boundary Layer ($SBL$) close to the surface, and the Residual Layer ($RL$) that contains features from the previous day's $ML$ and is just above the $SBL$.

Knowledge of the turbulent processes in the $CBL$ is important in diverse studies, mainly for atmospheric modeling and pollutant dispersion, since turbulent mixing can be considered as the primary process by which aerosol particles and other scalars are transported vertically in atmosphere. Because turbulent processes are treated as nondeterministic, they are characterized and described by their statistical properties (high order statistical moments). When applied to atmospheric studies such analysis provide information about the field of turbulent fluctuation, as well as, a description of the mixing process in the $PBL$ (Pal et al., 2010).

Anemometer towers have been widely applied in studies about turbulence (e.g., Kaimal and Gaynor, 1983; van Ulden and Wieringa, 1996), however the limited vertical range of these equipment restrict the analysis to regions close to surface. Aircraft have also been used in atmospheric turbulence studies (e.g., Lenschow et al., 1980; Williams and Hacker, 1992; Lenschow et al., 1994; Albrecht et al., 1995; Stull et al., 1997; Andrews et al., 2004; Vogelmann et al., 2012), nevertheless their short time window limits the analysis. In this scenario, systems with high spatial and temporal resolution and enough range are necessary in order to provide more detailed results along the day throughout the whole thickness of the $PBL$.

In the last decades, lidar systems have been increasingly applied in this kind of study due to their large vertical range, high data acquisition rate and capability to detect several observed quantities such as vertical wind velocity [Doppler lidar] (e.g. Lenschow et al., 2000; Lothon et al., 2006; O'Connor et al., 2010), water vapor [Raman lidar and DIAL] (e.g. Wulfmeyer, 1999; Kiemle et al., 2007; Wulfmeyer et al., 2010; Turner et al., 2014; Muppa et al., 2015), temperature [rotational Raman lidar] (e.g. Behrendt et al., 2015) and aerosol [elastic lidar] (e.g. Pal et al., 2010; McNicholas et al., 2015). This allows the observation of a wide range of atmospheric processes. For example, Pal et al. (2010) demonstrated how the statistical analyses obtained from high-order moments of elastic lidar can provide information about aerosol plume dynamics in the $PBL$ region. In addition, when different lidar systems operate synergistically, as for example in

Engelmann et al. (2008), who combined elastic and Doppler lidar data, it is possible to identify very
complex variables such as vertical particle flux.
Different works (Ansmann et al., 2010; O'Connor et al., 2010) have evidenced the feasibility for
characterizing the *PBL* turbulence by *DL*. Pal et al. (2010) have shown the feasibility for retrieving
information on the *PBL* turbulence from high high-order moments of elastic lidar operating at 1064. Such
approaches are even more attractive when considering facilities of networks, e. g. European Aerosol
Research Lidar NETwork (EARLINET) (Pappalardo et al., 2014), Microwave Radiometer Network
(MWRNET) (Rose et al., 2005; Caumont et al., 2016) and ACTRIS CLOUDNET (Illingworth et al., 2007).
For these reasons, and having in mind the wide spread of elastic lidar systems operated at other wavelengths,
like 532 nm or 355 nm, it would be worthy test the feasibility of these other wavelengths in the
characterization of the *PBL* turbulent behavior.
The use of simple techniques, applied to the aforementioned remote systems provide robust and similar
information on the *PBL* height (*PBLH*) during the convective period (see for example Moreira et al, 2018),
or a complementary information when the *CBL* is substituted by the presence of the *SBL* and the *RL*
(Moreira et al., in preparation). Thus, the combination of information obtained from the active remote
sensing systems, *DL* and *EL*, acquired with a temporal resolution close to 1 s, and that provided by *MWR*
can provide a detailed understanding about different features of the *PBL*, like structure (*CBL* versus *SBL*
and *RL*), height of the layers, rate of growth of the *PBLH* and turbulence.
In this study we show the feasibility of obtaining a clear insight on the *PBL* behavior using a combination
of active and passive remote sensing systems (Elastic Lidar [*EL*], Doppler Lidar [*DL*] and Microwave
Radiometer [*MWR*]) acquired during the SLOPE-I campaign, held at IISTA-CEAMA (Andalusian Institute
for Earth System Research, Granada, Spain) from May to August 2016. One of the goals is to show the
feasibility of using *EL* at 532 nm, considering the widespread use of lidar systems based on laser emission
at this wavelength in different coordinated networks, like as EARLINET (Pappalardo et al., 2014) and
LALINET – Latin American LIdar Network (Guerrero-Rascado et al., 2016). In addition, this study shows
the variety of application that can be done with EARLINET data applying some simple changes in the data
acquisition procedures.
This paper is organized as follows. Description of the experimental site and the equipment setup are
presented in Section 2. The methodologies applied are introduced in Section 3. Section 4 presents the results
of the analyses using the different methodologies. Finally, conclusions are summarized in Section 5.

**2 Experimental site and instrumentation**
The SLOPE-I (Sierra nevada Lidar aerOsol Profiling Experiment) campaign was performed from May to
September 2016 in South-Eastern Spain in the framework of the European Research Infrastructure for the
observation of Aerosol, Clouds, and Trace gases (ACTRIS). The main objective of this campaign was to
perform a closure study by comparing remote sensing system retrievals of atmospheric aerosol properties,
using remote systems operating at the Andalusian Institute of Earth System Research (IISTA-CEAMA)
and in-situ measurements operating at different altitudes in the Northern slope of Sierra Nevada, around 20
km away from IISTA-CEAMA (Bedoya-Velásquezet al., 2018; Román et al., 2018). The IISTA-CEAMA
station is part of EARLINET (Pappalardo et al, 2014) since 2005 and at present is an ACTRIS station
(http://actris2.nilu.no/). The research facilities are located at Granada, a medium size city in Southeastern
Spain (Granada, 37.16°N, 3.61°W, 680 m a.s.l.), surrounded by mountains and with Mediterranean-
continental climate conditions that are responsible for cool winters and hot summers. Rain is scarce,
especially from late spring to early autumn. Granada is affected by different kind of aerosol particles locally
originated and medium-long range transported from Europe, Africa and North America (Lyamani et al.,
2006; Guerrero-Rascado et al., 2008, 2009; Titos et al., 2012; Navas-Guzmán et al., 2013; Valenzuela et
al., 2014, Ortiz-Amezcua et al, 2014, 2017).
MULHACÉN is a biaxial ground-based Raman lidar system operated at IISTA-CEAMA in the frame of
EARLINET research network. This system operates with a pulsed Nd:YAG laser, frequency doubled and
tripled by Potassium Dideuterium Phosphate crystals, emitting at wavelengths of 355, 532 and 1064 nm
with output energies per pulse of 60, 65 and 110 mJ, respectively. MULHACÉN operates with three elastic
channels: 355, 532 (parallel and perpendicular polarization) and 1064 nm and three Raman-shifted
channels: 387 (from $N_2$), 408 (from $H_2O$) and 607 nm (from $N_2$). MULHACÉN's overlap is complete at
90% between 520 and 820 m a.g.l. for all the wavelengths, reaching full overlap around 1220 m a.g.l.
(Navas-Guzmán et al ., 2011; Guerrero-Rascado et al. 2010). Calibration of the depolarization capabilities
is done following Bravo-Aranda et al. (2013). This system was operated with a temporal and spatial
resolution of 2 s and 7.5 m, respectively. More details can be found at Guerrero-Rascado et al. (2008, 2009).
The Doppler lidar (Halo Photonics, model Stream Line XR) is also operated at IISTA-CEAMA. This
system works in continuous and automatic mode from May 2016. It operates at 1.5 µm with pulse energy
and repetition rate of 100 µJ and 15 KHz, respectively. This system records the backscattered signal with a
range resolution of 30 m in 300 range gates with the first range gate starting at 60 m from the instrument.
The telescope focus is set to approximately 800 m. The instrument was operated in vertical stare mode with
a temporal resolution of 2 s.
Furthermore, we operated the ground-based passive microwave radiometer (RPG-HATPRO G2,
Radiometer Physics GmbH), which is member of the MWRnet [http://cetemps.aquila.infn.it/mwrnet/]. This
system operates in automatic and continuous mode at IISTA-CEAMA since November 2011. The
microwave radiometer (MWR) measures the sky brightness temperature with a radiometric resolution
between 0.3 and 0.4 K root mean square error at 1 s integration time, using direct detection receivers within
two bands: K-band (water vapor – frequencies: 22.24 GHz, 23.04 GHz, 23.84 GHz, 25.44 GHz, 26.24 GHz,
27.84 GHz, 31.4 GHz) and V-band (oxygen – frequencies: 51.26 GHz, 52.28 GHz, 53.86 GHz, 54.94 GHz,
56.66 GHz, 57.3 GHz, 58.0 GHz). From these bands is possible to obtain profiles of water vapor and
temperature, respectively, by inversion algorithms described in Rose et al. (2005). The range resolution of
these profiles vary between 10 and 200 m in the first 2 km and between 200 and 1000 m in the layer between
2 and 10 km (Navas-Guzmán et al., 2014).
The meteorological sensor (HMP60, Vaisala) is used to register the air surface temperature and surface
relative humidity, with a temporal resolution of 1 minute. Relative humidity is monitored with an accuracy
of ± 3%, and air surface temperature is acquired with an accuracy and precision of 0.6º C and 0.01º C,
respectively.
A CM-11 pyranometer manufactured by Kipp&Zonen (Delft, The Netherlands) is also installed in the
ground-based station. This equipment measures the shortwave (SW) solar global horizontal irradiance data
(305–2800 nm). The CM-11 pyranometer complies with the specifications for the first-class WMO (World
Meteorological Organization) classification of this instrument (resolution better than $\pm5$ Wm$^{-2}$), and the
calibration factor stability has been periodically checked against a reference CM-11 pyranometer (Antón
et. al, 2012).
**3 Methodology**
**3.1 MWR data analysis**
The MWR data are analyzed combining two algorithms, Parcel Method [$PM$] (Holzworth, 1964) and
Temperature Gradient Method [$TGM$] (Coen, 2014), in order to estimate the $PBL$ Height ($PBLH_{MWR}$) in
convective and stable situations, respectively. The different situations are discriminated by comparing the
surface potential temperature ($\theta(z_0)$) with the corresponding vertical profile of $\theta(z)$ up to 5 km. Those
cases where all the points in the vertical profile have values larger than $\theta(z_0)$ are labeled as stable, and
$TGM$ is applied. Otherwise the situation is labeled as unstable and the $PM$ is applied. The vertical profile
of $\theta(z)$ is obtained from the vertical profile of $T(z)$ using the following equation (Stull, 2011):
$$\theta(z) = T(z) + 0.0098 * z \quad (1)$$

where $T(z)$ is the temperature profile provided by $MWR$, $z$ is the height above the sea level, and 0.0098
K/m is the dry adiabatic temperature gradient. A meteorological station co-located with the $MWR$ is used
to detect the surface temperature [$T(z_0)$]. In order to reduce the noise, $\theta(z)$ profiles were averaged
providing a $PBLH_{MWR}$ value at 30 minutes intervals. This methodology of $PBLH$ detection was selected as
the reference due to the results obtained during a performed intercomparison campaign between $MWR$ and
radiosonde data, where twenty-three radiosondes were launched. High correlations were found between
$PBLH$ retrievals provided by both instruments in stable and unstable cases. Further details are given by
Moreira et al. (2018a).
**3.2 Lidar retrieval of the PBLH.**
The simple processing of $DL$ and $EL$ data allows the estimation of the $CBL$ height. Moreira et al. (2018),
have discussed this issue in depth, while Moreira et al. (in preparation) have exploited the complementarity
of the data obtained from distinct remote sensing systems in order to distinguish the sublayers during the
period when the *SBL* and *RL* substitute the *CBL*, as well as, in complex situations, like as, presence of dust
layers.
The *PBLH* obtained from *DL* data ($PBLH_{Doppler}$) is estimated from variance threshold method. In this
method the $PBLH_{Doppler}$ is attributed to height where the variance of vertical wind speed ($\sigma_w^2$) is lower than
a determinate threshold, which was adopted as 0.16 m²/s² (Moreira et al., 2018). For the $PBLH_{Doppler}$
calculations was selected a time interval of 30 minutes. In concerning the *PBLH* obtained from *EL*
($PBLH_{Elastic}$), the variance method is applied. Such method assumes the maximum of the variance of
Range Corrected Signal ($\sigma_{RCS}^2$) as $PBLH_{Elastic}$ (Moreira et al., 2015). The $\sigma_{RCS}^2$ is obtained from a time
interval of 30 minutes.
**3.3 Lidar turbulence analysis**
Both lidar systems, *DL* and *EL*, gathered data [$q(z,t)$] with a temporal resolution of 2 seconds. Then, the
data are averaged in 1-hour packages, from which the mean value is extracted [$\bar{q}(z)$]. Such mean value is
subtracted from each $q(z,t)$ profile in order to estimate the vertical profile of the fluctuation for the
measured variable [$q'(z,t)$] (i.e. vertical velocity for the *DL*):
$$q'(z,t) = q(z,t) - \bar{q}(z) \quad (2)$$

Then, from $q'(z,t)$ is possible to obtain the high-order moments (variance ($\sigma^2$), skewness ($S$) and kurtosis
($K$)), as well as, the integral time scale ($\tau$ - which is the time over which the turbulent process are highly
correlated to itself) as shown in Table 1. These variables can also be obtained from the following
autocovariance function, $M_{ij}$:
$$M_{ij} = \int_0^{t_f} [q'(z,t)]^i [q'(z,t+t_f)]^j dt \quad (3)$$

where $t_f$ is the final time, $i$ and $j$ indicate the order of autocovariance function.
However, it is necessary to considerer that the acquired real data contain instrumental noise, $\varepsilon(z)$.
Therefore, the equation 3 can be rewritten as:
$$M_{ij} = \int_0^{\tau} [q(z,t) + \varepsilon(z,t)]^i [q(z,t+\tau) + \varepsilon(z,t+\tau)]^j dt \quad (4)$$

The autocovariance function of a time series with zero lag results in the sum of the variances of the
atmospheric variable and its $\varepsilon(z)$. Nevertheless, atmospheric fluctuations are correlated in time, but the
$\varepsilon(z)$ is random and uncorrelated with the atmospheric signal. Consequently, the noise is only associated
with lag 0 (Fig. 1). Based on this concept Lenschow et al. (2000) suggested to obtain the corrected
autocovariance function, $M_{11}(\to 0)$, from two methods, namely first lag correction or -2/3 law correction.
In the first method, $M_{11}(\to 0)$ is obtained directly by the subtraction of lag 0, $\Delta M_{11}(0)$, from the
autocovariance function, $M_{11}(0)$. In the second method $M_{11}(\to 0)$ is generated by the extrapolation of
$M_{11}(0)$ at firsts nonzero lags back to lag zero (-2/3 law correction). The extrapolation can be performed
using the inertial subrange hypothesis, which is described by the following equation (Monin and Yaglom,
214 1979):

$$M_{11}(\to 0) = \overline{q'^2(z,t)} + Ct^{2/3} (5)$$

where C represents a parameter of turbulent eddy dissipation rate. The high-order moments and $\tau$
corrections and errors are shown in Table 1 (columns 2 and 3, respectively).
The same procedure of analysis is applied in studies with $DL$ and $EL$, being the main difference the tracer
used by each system, which are the fluctuation of vertical wind speed ($w'$) for $DL$ and aerosol number
density ($N'$) for $EL$. $DL$ provides $w(z,t)$ directly, and therefore the procedure described in Figure 2 can be
directly applied. Thus, the two corrections described above are applied separately and finally $\tau$ and high-
order moments with and without corrections can be estimated.
On the other hand, the $EL$ does not provide $N(z,t)$ directly. Under some restrictions, it is possible to ignore
the particle hygroscopic growth and to assume that the vertical distribution of aerosol type does not changes
with time, and to adopt the following relation (Pal et al., 2010):

$$\beta_{par}(z,t) \approx N(z,t)Y(z) \Rightarrow \beta'_{par}(z,t) = N'(z,t) (6)$$

where $\beta_{par}$ and $\beta'_{par}$ represent the particle backscatter coefficient and its fluctuation, respectively, and
$Y(z)$ does not depend on time.
Considering the lidar equation:

$$P_\lambda(z) = P_0 \frac{ct_d}{2} AO(z) \frac{\beta_\lambda(z)}{z^2} e^{-2\int_0^z \alpha_\lambda(z'dz')} (7)$$

where $P_\lambda(z)$ is the signal returned from distance $z$ at time $t$, $z$ is the distance [m] from the lidar of the
volume investigated in the atmosphere, $P_0$ is the power of the emitted laser pulse, $c$ is the light speed [m/s],
$t_d$ is the duration of laser pulse [ns], $A$ is the area [m²] of telescope cross section, $O(z)$ is the overlap
function, $\alpha_\lambda(z)$ is the total extinction coefficient (due to atmospheric particles and molecules) [(km)⁻¹] at
distance $z$, $\beta_\lambda(z)$ is the total backscatter coefficient (due to atmospheric particles and molecules) [(km·sr)⁻
¹] at distance $z$ and the subscript $\lambda$ represents the wavelength. The two path transmittance term related to
$\alpha(z)$ is considered as nearly negligible at 1064 nm (Pal et al., 2010). Thus, it is possible to affirm that:

$$RCS_{1064}(z) = P(z)_{1064}.z^2 \cong G.\beta_{1064}(z) (8)$$

and consequently:

$$RCS'_{1064}(z,t) \cong \beta'_{1064}(z,t) = \beta'_{par}(z,t) = N'(z,t) (9)$$

where $RCS_{1064}$ and $RCS'_{1064}$ are the range corrected signal and its fluctuation, respectively, $G$ is a constant
and the subscripts represent the wavelength.
In this way, Pal et al. (2010) have shown the feasibility of using $EL$ operating at 1064 nm for describing
the atmospheric turbulence. However, having in mind the more extended use of lidar systems based on
laser emission at 532 nm in different coordinated networks, e.g., in EARLINET and LALINET around 76%
and 45% of the systems include the wavelength of 1064 nm, while 95% of the EARLINET systems and
73% of the LALINET systems operate systems that include the wavelength 532 nm (Guerrero-Rascado et
al., 2016), in this study we evaluate using $RCS_{532}$ fluctuations to determine turbulence following the
procedure described in Figure 3. This $EL$ methodology is very similar to that described earlier for $DL$.
we perform the validation of the $RCS_{532}$ in analyses about turbulence using $EL$, following the procedure
described in Figure 3, which is basically the same methodology described earlier for $DL$.
**4 Results**
**4.1 Error Analysis**
The influence of random error in noisy observations rapidly grows for higher-order moments (i.e., the
influence of random noise is much larger for the fourth-order moment than for the third-order moment).
Therefore, the first step, in order to ascertain the applied methodology and our data quality, we performed
the error treatment of $DL$ data as described in Figure 2. For the $DL$ analysis we selected the period 08-09
UTC of 19[th] May, the same day that will be presented in Case Study 1. This day is characterized by a well-
defined PBL.
Figure 4 illustrates the autocovariance function, generated from $w'$, at three different heights. As mentioned
before, the lag 0 is contaminated by noise ($\varepsilon$), and thus the impact of the $\varepsilon$ increases together with height,
mainly above $PBLH_{MWR}$ (1100 m a.g.l. in our example).
Figure 5-A illustrates the comparison between integral time scale ($\tau_{w'}$) without correction and the two
corrections cited in section 3.2. Except for the first height-bins, below the $PBLH_{MWR}$ the profiles have little
differences, as well as small errors bars. Above the $PBLH_{MWR}$ the first lag correction presents higher
differences in relation to the other profiles at around 1350 m.
Figures 5-B and 5-C show the comparison of variance ($\sigma_{w'}^2$) and skewness ($S_{w'}$), respectively, with and
without corrections. The profiles corrected by -2/3 law do not present significant differences in comparison
to uncorrected profiles. On the other hand, the profiles corrected by the first lag correction have slight
differences below the $PBLH_{MWR}$, mainly the $\sigma_{w'}^2$, ($S_{w'}$only in the first 50 m)**.** Therefore, considering high
Signal-to-Noise Ratio ($SNR$) conditions, although the presence of $\varepsilon$ can change slightly the value of high
order moments, it is not enough to distort the observed phenomena as shown by the impact of the corrections
applied.
For $EL$ we use the same procedure for the correction and error analysis that we apply to the $DL$ data. The
same day was chosen (19[th] May), however the period selected is between 12 and 13 UTC, due to the
incomplete overlap of MULHACÉN.
In this sense, we studied the influence of noise at two wavelengths: 1064 nm, that has been previously
analyzed by Pal et al. (2010) as presented in the section 2 and adopted as reference (considering the rather
low impact of molecular signal and the two ways transmittance shown in 9) and 532 nm, just in order to
check the feasibility of this wavelength for turbulence studies considering its widespread use in observation
networks (Pappalardo et al., 2014; Guerrero-Rascado et al., 2016). Figures 6 and 7 shows the
autocovariance function, obtained from $RCS'_{1064}$ and $RCS'_{532}$, respectively, at three distinct heights. As
expected, $\varepsilon$ increases with range, principally above the $PBLH_{MWR}$. However, the wavelength 532 nm is
more influenced by the noise, what can be verified by the higher peak at lag 0 in figure 7, in comparison
with peaks at same lag in figure 6.
Although the level of influence of $\varepsilon$ in each wavelength depends on the $SNR$ of them (which is associated
to technical factors such as laser output power, filters, type of detectors), considering the proposed
methodology, to evaluate the composition of each wavelength is also important. The large contribution of
$\beta_{Molecular}^{532}$ to the total $\beta$ at 532 nm in comparison with the behavior at 1064 nm, can influence the results
obtained from such wavelength, because our methodology is based on the use of $\beta'_{Aerosol}$. In addition, the
larger extinction (due to both aerosol particles and molecules) at 532 nm produces a lower two-way
transmittance, resulting in the reduction of the $SNR$ values at this wavelength. As we used Elastic lidar
technique, we could not calculate aerosol extinction profiles, but an estimation of these transmittances was
done on the basis of Klett method (Klett, 1985). With this method, a constant lidar ratio value was
constrained for each profile using the AOD derived from a collocated AERONET Sun-photometer
(Guerrero-Rascado et al., 2008). Using these constrained lidar ratios, the transmittances were calculated
together with aerosol backscatter profiles, integrated up to 2.5 km. The estimated two-way transmittance
was 0.85 for the case analyzed in this subsection (19th May).
Figures 8-A, 8-B, 8-C and 8-D show the vertical profiles of $\tau_{RCS'}$, $\sigma_{RCS'}^2$, $S_{RCS'}$ and kurtosis ($K_{RCS'}$),
respectively, obtained at 1064 nm, with and without the corrections described in section 3.2. In general, the
corrections do not affect the profiles generated from 1064 nm data in a significant way, so that, the higher
influence of corrections is observed in the $K_{RCS'}$ profile, which is underestimated in some regions. In the
figures 9-A, 9-B, 9-C and 9-D we show same high order moments calculated from 532 nm data. As the
complexity of moments increases, it is possible to observe the larger influence of the corrections, due to
propagation of noise. Nonetheless, the application of the corrections, mainly first lag correction, make these
profiles very similar to those generated from the wavelength 1064 nm, so that the same phenomena can be
observed in both.
Therefore, in spite of the larger attenuation expected at 532 nm wavelength, which reduces the $SNR$ of the
profiles in comparison with 1064 nm, the application of the proposed corrections, mainly the first lag,
reduces significantly such influence and enable the observation of the same phenomena detected in the
high-order moments obtained from 1064 nm. Consequently, the wavelength 532 nm will be applied in the
analysis presented in section 4.2. The first lag correction was adopted as default due to it generates more
relevant results in comparison with the -2/3 law correction, providing a more careful analysis.

**4.2 Case studies**

In this section we present two study cases, in order to show how the products indicated in table 2 can
provide a detailed description about the turbulence in the $PBL$. The first case represents a typical day with
a clear sky situation. The second case corresponds to a more complex situation, where there is presence of
clouds and Saharan mineral dust layers.
**4.2.1 Case study I: clear sky situation**
In this case study we use measurements gathered with $DL$, $MWR$ and pyranometer during 24 hours. The
$EL$ was operated under operator-supervised mode between 08:20 to 18:00 UTC.
Figure 10 (A) shows the integral time scale obtained from $DL$ data ($\tau_{w\prime}$). The gray area represents the region
where it is not possible to analyze the turbulent process from our $DL$ data, either because of the low $SNR$
values, which results in null values of the $\tau_{w\prime}$, or due to regions where the $\tau_{w\prime}$ is not null, but it is lower
than the acquisition time of the $DL$. However, the gray area is located almost entirely above the $PBLH_{MWR}$
(white stars).
The $\sigma_{w\prime}^2$ has low values during the entire period when the $SBL$ is present (Figure 10-B). Nevertheless, as air
temperature begins to increase (around 07:00 UTC), the $\sigma_{w\prime}^2$ increases together, as well as, the $PBLH_{MWR}$.
The $\sigma_{w\prime}^2$ reaches its maximum values in the middle of the day, when we also observe the maximum values
of air temperature and $PBLH_{MWR}$.. The combination of $\sigma_{w\prime}^2$ and $PBLH_{MWR}$ provides us a better
comprehension about the $PBLH$ growth speed, so that, in the moments where high values of $\sigma_{w\prime}^2$ are
observed, it means higher values of Turbulent Kinetic Energy ($TKE$), which favor the fast ascension of
$PBLH$.
The skewness of $w\prime$ ($S_{w\prime}$) is shown in Figure 11-C. The $S_{w\prime}$ describes the distribution of the turbulent
velocities. Thus positive $S_{w\prime}$ implies strong but narrow updrafts surrounded by weaker but more widespread
downdrafts, and vice versa for negative $S_{w\prime}$. Consequently, positive values (red regions) correspond with a
surface-heating-driven boundary layer, while negative (blue regions) ones are associated to cloud-top long-
wave radiative cooling. During the stable period, there is predominance of low absolute values of $S_{w\prime}$.
Nevertheless, as air temperature increases (transition from stable to unstable period), $S_{w\prime}$ values begin to
become larger. Air temperature begins to decrease around 18:00 UTC, and there is a reduction of $S_{w\prime}$, so
that, the generation rate of convective turbulence decreases. Therefore, the turbulence cannot be maintained
against dissipation, then the $CBL$ becomes a $SBL$ covered by the $RL$. Thus, the reduction observed in the
$PBLH_{MWR}$ is due to the detection of $SBL$ height.
Figure 10-D shows the values of net surface radiation ($R_n$) that are estimated from solar global irradiance
values using the seasonal model described in Alados et al. (2003). The negative values of $R_n$ are
concentrated in the stable region. The $R_n$ begins to increase around 06:00 UTC and reaches its maximum
in the middle of the day. Comparing figures 8-C and 8-D, we can observe similarity among the behavior of
$S_{w\prime}$ and $R_n$, so that, the joint analysis of these variables reinforce the characterization of this $PBL$ as surface-
heating-driven $CBL$.
Figure 10-E presents the values of surface air temperature and surface relative humidity ($RH$). Air surface
temperature has a pattern of increase and decrease similar to observed in $R_n$ and $S_{w\prime}$. On the other hand,
$RH$ is inversely correlated with temperature.
Figure 11 shows the $RCS_{532}$ profile obtained from 08:00 to 18:00 UTC. At the beginning of the
measurement period (08:20 to 10:00 UTC) it is possible to observe the presence of a thin residual layer
(around 2000 m a.s.l.), and later from 13:00 to 18:00 UTC it is evident a lofted aerosol layer. In this picture
there are the $PBLH_{MWR}$ (pink stars), the $PBLH_{Doppler}$ (blue stars), obtained from the maximum of $\sigma_{w\prime}^2$,
(Moreira et al., 2018a), and the $PBLH_{Elastic}$ (black stars), obtained from the maximum of $\sigma_{RCS\prime}^2$ (Moreira
et al., 2015). In the initial part of measurement, all profiles have similar behavior. However due to distinct
$PBLH$ definition and tracer applied by each one, the differences increase as $CBL$ becomes more complex,
e.g. the presence of lofted aerosol layer at 14 UTC. The joint observation of the results provided by these
three methods can provide us information about the sublayers in the $PBL$, both in convective and stable
situations. Due to low variability of $PBLH$, the period between 13:00 and 14:00 UTC has been selected to
be analyzed from the high order moments.
Figure 12 presents the statistical moments generated from $RCS\prime$ of wavelength 532 nm, which were obtained
from 13:00 and 14:00 UTC. The red line in all graphics represent the $PBLH_{Elastic}$ (2200 m a.s.l.) and the
blue one the average value of $PBLH_{MWR}$ (2250 m a.s.l.), both obtained between 13 and 14 UTC.
Due to presence of a decoupled aerosol layer at 13:30, the average values of $PBLH_{Elastic}$ and $PBLH_{MWR}$
have a difference of around 500 m. The $\sigma_{RCS\prime}^2$ has small and practically constant values between 1000 and
1400m, evidencing the homogeneity of aerosol distribution in this region. From 1400 m the value of $\sigma_{RCS\prime}^2$
begins to increase, reaching a positive peak at $PBLH_{MWR}$ , which represents the Entrainment Zone (region
characterized by an intense mixing between air parcels coming from CBL and Free Troposphere (FT),
causing a high variation in aerosol concentration). The $PBLH_{Elastic}$ observed at approximately 2900 m
demonstrate an inherent difficulty of variance method to detect the PBLH in the presence of several aerosol
layers (Kovalev and Eichinger, 2004).  Above $PBLH_{Elastic}$ the values of $\sigma_{RCS\prime}^2$ decrease slowly due to
location of the lofted aerosol around 2500 m. However, above this aerosol layer the value of $\sigma_{RCS\prime}^2$ is
reduced to zero, indicating a large homogeneity in aerosol distribution at this region, what is expected,
because the aerosol concentration at the $FT$ is negligible in this case. The integral time scale obtained from
$RCS\prime$ ($\tau_{RCS\prime}$) has values higher than $EL$ time acquisition throughout the $CBL$, evidencing the feasibility for
studying turbulence using this elastic lidar configuration. The skewness values obtained from $RCS\prime$ ($S_{RCS\prime}$)
give us information about aerosol motion. The positive values of $S_{RCS\prime}$ observed in the lowest part of profile
and above the $PBLH_{Elastic}$ represents the updrafts aerosol layers. The negative values of $S_{RCS\prime}$ indicates the
region with low aerosol concentration due to clean air coming from $FT$. This movement of ascension of
aerosol layers and descent of clean air with zero value of $S_{RCS\prime}$ at $PBLH$ (characteristic of the $CBL$ growing)
was also detected by Pal et al. (2010) and McNicholas et al. (2014). The kurtosis of $RCS\prime$ ($K_{RCS\prime}$) determines
the level of mixing at different heights. There are values of $K_{RCS\prime}$ larger than 3 in the lowest part of profile
and around 2500 m, showing a peaked distribution in this region. On other hand, values of $K_{RCS'}$ lower than
3 are observed close to the $PBLH_{Elastic}$, therefore this region has a well-mixed $CBL$ regime. Pal et al. (2010)
and McNicholas et al. (2014) also detected this feature in the region nearby the $PBLH$. In figure 13 are
shown the high-order moments obtained at the same period described above, however from the 1064 nm
data (our reference wavelength). It is possible to observe a similarity between the profiles obtained from
each wavelength, so that, the same phenomena observed in the profiles generated from 532 nm and
described above, also are detected in the profiles obtained from the reference wavelength.
The results provided by $DL$, pyranometer and $MWR$ data agree with the results observed in figures 12 and
13. In the same way, the analysis of high order moments of $RCS'$ fully agree with the information in Figure
10. Thus, the large values of $S_{RCS'}$ and $K_{RCS'}$ detected around 2500 m a.s.l, where we can see a lofted aerosol
layer, suggest the ascent of an aerosol layer and presence of a peaked distribution, respectively.

### 397  4.2.2 Case study: dusty and cloudy scenario

In this case study measurements with $DL$, $MWR$ and pyranometer expand during 24 hours, while $EL$ data
are collected from 09:00 to 16:00 UTC.
Figure 14-A shows $\tau_{w'}$. Outside the period 13:00 to 17:00 UTC, the greatest part of grey area is situated
above the $PBLH_{MWR}$ (white stars), thus $DL$ time acquisition is enough to perform studies about turbulence
in this case.
$\sigma_{w'}^2$ has values close to zero during all the stable period (Figure 14-B). However, when air temperature
begins to increase (around 06:00 UTC), the $\sigma_{w'}^2$ also increases and reaches its maximum in the middle of
the day. The higher values of $PBL$ growth speed are observed in the moments where $\sigma_{w'}^2$ reaches its
maximum values. In the late afternoon, as air temperature decrease, the values of $\sigma_{w'}^2$ (and consequently
the $TKE$) decrease gradually, until reach the minimum value associated to the $SBL$. Figure 14-C shows the
profiles of $S_{w'}$. The main features of this case are: the low values of $S_{w'}$, the slow increase and ascension
of positive $S_{w'}$ values and the predominance of negative $S_{w'}$ values from 12:00 to 13:00 UTC. The first two
features are likely due to the presence of the intense Saharan dust layer (Figure 15), which reduces the
transmission of solar irradiance, and consequently the absorption of solar irradiance at the surface,
generating weak convective process. From figure 16 we can observe the presence of both middle altitude
clouds and very intense dust layers from 12:00 to 15:00 UTC. Such combination contributes to the intense
negative values of $S_{w'}$ observed in this period until around 2 km, because, as mentioned previously, $S_{w'}$ is
directly associated with the direction of turbulent movements, which during this period can be characterized
as cloud-top long-wave radiative cooling (Ansmann et al., 2010).
The influence of Saharan dust layer can also be evidenced on the $R_n$ pattern (Figure 14-D), which maintains
negative values until 12:00 UTC and reaches a low maximum value (around 200 W/m²). The observation
of $S_{w'}$ and $R_n$ between 12:00 and 14:00, as well as, the presence of clouds and geometrically thick dust
layers during this same period, reinforces the hypothesis of a case of the cloud-top long-wave radiative
cooling in the $CBL$. Air surface temperature and $RH$ (Figure 14-E) present the same correlation and anti-
correlation (respectively) observed in the earlier case study, where the maximum of air surface temperature
and the minimum of $RH$ are detected in coincidence with the maximum daily value of $PBLH_{MWR}$.
As mentioned before, Figure 15 shows the $RCS$ profile obtained from 09:00 to 16:00 UTC in a complex
situation, with presence of decoupled dust layer (around 3800 m a.s.l.) from 09:00 to 12:00 UTC and the
presence of both middle altitude clouds and very intense dust layers (around 3500 m a.s.l.) from 11:30 to
16:00 UTC. The pink, black and blue stars represent the $PBLH_{MWR}$, $PBLH_{Doppler}$ and $PBLH_{Elastic}$
respectively. Due to the presence of dusty layers and clouds, the difference between the methods is more
evident, mainly of the $PBLH_{Elastic}$, which uses the aerosol as tracers. This method only produces results
close to the others at 15 UTC, when dust layer is mixed with the $CBL$.
Figure 16 illustrates the statistical moments of $RCS'$ of 532 nm wavelength obtained from 11:00 to 12:00
UTC. The $\sigma^2_{RCS'}$ profile presents several peaks due to the presence of distinct aerosol sublayers. The first
peak is coincident with the value of $PBLH_{MWR}$. The value of $PBLH_{elastic}$, is coincident with the base of
the dust layer. This difficulty to detect the $PBLH$ in presence of several aerosol layers is inherent to the
variance method (Kovalev and Eichinger, 2004). However, the joint observation of $PBLH_{MWR}$ and
$PBLH_{elastic}$, enable us to characterize and distinguish the several sublayers. The values of $\tau_{RCS'}$ are higher
than $EL$ acquisition time all along the $PBL$, evidencing the feasibility of $EL$ time acquisition for studying
the turbulence of $PBL$ in this case. The $S_{RCS'}$ profile has several positive values, due to the large number of
aerosol sublayers that are present. The characteristic inflection point of $S_{RCS'}$ is observed in coincidence
with the $PBLH_{MWR}$, that confirming the agreement between this point and the $PBLH$. From the analysis of
$S_{RCS'}$ and $S_{w'}$ is possible to justify this phenomena from the mixing process demonstrated in the earlier case
study. The $K_{RCS'}$ has predominantly values lower than 3 below 2500 m, thus shown how this region is well
mixed as can see in Figure 16. Values of $K_{RCS'}$ larger than 3 are observed in the highest part of profile, where
the dust layer is located.
In order to show the feasibility of 532 nm wavelength, in the figure 17 are presented the high-order moments
obtained between 11-12 UTC from 1064 nm wavelength data. Although the error of $\sigma^2_{RCS'}$ obtained from
532 nm (pink shadow) is considerably higher than the error of same variable obtained from 1064 nm, all
profiles are very similar, so that, the same phenomena can be observed in both graphics (figure 16 and 17).
Figure 18 shows the $RCS'$ 532 nm wavelength high-order moments obtained from 12:00 and 13:00 in
presence of cloud cover. The method based on maximum of $\sigma^2_{RCS'}$ locates the $PBLH_{Elastic}$ at the cloud base,
due to the high variance of $RCS'$ generated by the clouds. $\tau_{RCS'}$ presents values larger than $EL$ time
acquisition, therefore this configuration enable us to study turbulence by $EL$ analyses. $S_{RCS'}$ has few peaks,
due to the mixing between $CBL$ and dust layer, generating a more homogenous layer. The highest values
of $S_{RCS'}$ are observed in regions where there are clouds, and the negative ones (between 3500 and 4000 m)
occur due to presence of air from $FT$ between the two aerosol layers (Figure 15). The inflection point of
$S_{RCS'}$ profile is observed in $PBLH_{MWR}$ region. $K_{RCS'}$ profile has low values in most of the $PBL$, demonstrating
the high level of mixing during this period, where dust layer and $PBL$ are combined. The higher values of
$K_{RCS'}$ are observed in the region of clouds. In the same way of the previous analysis, the high-order moments
of the period mentioned above were calculated for the wavelength of 1064 nm (figure 19). Although there
are some differences in the absolute values of some profiles, the high-order moments generated using 1064
and 532 nm have similar profiles, so that, the same phenomena can be observed, demonstrating the viability
of 532 nm wavelength in the proposed methodology.

## 5 Conclusions

In this paper we perform an analysis about the $PBL$ turbulent features from three different types of remote
sensing systems ($DL$, $EL$and $MWR$) and surface sensors during SLOPE-I campaign. We applied two kind
of corrections to the lidar data: first lag and -2/3 corrections. The corrected $DL$ statistical moments showed
little variation with respect to the uncorrected profiles, denoting a rather low influence of the noise in high
$SNR$ conditions. The $EL$ high-order moments were obtained from two wavelengths: 1064 nm, adopted as
reference, and 532 nm, in order to verify the viability to use the last one in turbulence analysis. From this
comparison, was possible to observe that the wavelength 532 nm is more affected by noise, in comparison
with 1064 nm, due to the large contribution of the molecular component and the reduced two-way
transmittance at that wavelength. However, the application of proposed corrections, mainly the first lag,
can reduce such influence, so that, the same phenomena can be observed in the high-order moments
provided from both wavelengths
The case studies present two kind of situations: well-defined PBL and a more complex situation with the
presence of Saharan dust layer and some clouds. In both cases was possible to identify the events describe
in table 2. The combined use of remote sensing systems shows how the results provided by the different
instruments can complement one each other, providing a detailed observation of some phenomena, mainly
in complex situations.
Therefore, this study shows the feasibility of the described methodology based on the combination of
remote sensing systems for retrieving a detailed picture on the $PBL$ turbulent features. In addition, the
feasibility of using the analyses of high order moments of the $RCS$ collected at 532 nm at a temporal
resolution of 2 s offers the possibility for using the proposed methodology in networks such as EARLINET
or LALINET with a reasonable additional effort.

**Acknowledgements**

This work was supported by the Andalusia Regional Government through project P12-RNM-2409, by the
Spanish AgenciaEstatal de Investigación, AEI, through projects CGL2016-81092-R and CGL2017-90884-
REDT. We acknowledge the financial support by the European Union's Horizon 2020 research and
innovation program through project ACTRIS-2 (grant agreement No 654109). The authors thankfully
acknowledge the FEDER program for the instrumentation used in this work and the University of Granada
that supported this study through the Excellence Units Program and "Plan Propio. Programa 9 Convocatoria
2013".

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

Table 1 – Variables applied to statistical analysis (Lenschow et al., 2000)

| | Without Correction | Correction | Error |
|---|---|---|---|
| **Integral Time Scale ($\tau$)** | $\displaystyle\int_0^\infty q'(t)dt$ | $\displaystyle\frac{1}{\overline{q'^2}}\int_{t\to 0}^\infty M_{11}(t)dt$ | $\displaystyle\tau.\sqrt{\frac{4\Delta M_{11}}{M_{11}(\to 0)}}$ |
| **Variance ($\sigma_{q'}^2$)** | $\displaystyle\frac{1}{T}\sum_{t=1}^T (q(t)-\overline{q})^2$ | $M_{11}(\to 0)$ | $\displaystyle q^2.\sqrt{\frac{4\Delta M_{11}}{M_{11}(\to 0)}}$ |
| **Skewness ($S$)** | $\displaystyle\frac{\overline{q^3}}{\sigma_q^3}$ | $\displaystyle\frac{M_{21}(\to 0)}{M_{11}^{3/2}(\to 0)}$ | $\displaystyle\frac{\Delta M_{21}}{\Delta M_{11}^{3/2}}$ |
| **Kurtosis ($K$)** | $\displaystyle\frac{\overline{q^4}}{\sigma_q^4}$ | $\displaystyle\frac{3M_{22}(\to 0)-2M_{31}(\to 0)-3\Delta M_{11}^2}{M_{11}^2(\to 0)}$ | $\displaystyle\frac{4\Delta M_{31}-3\Delta M_{22}-\Delta M_{11}^2}{\Delta M_{11}^2}$ |




Table 2 – Products and their respective meaning, provided by each system

| Product | System | Meaning |
|---|---|---|
| $\tau_{w'}(z)$ | Doppler lidar | Measurement in time of length of turbulent eddies |
| $\sigma_{w'}^2(z)$ | Doppler lidar | Turbulent Kinetic Energy |
| $S_{w'}(z)$ | Doppler lidar | Direction of turbulent movements |
| $PBLH_{Doppler}$ | Doppler lidar | Top of CBL obtained from variance threshold method |
| $\tau_{RCS'}(z)$ | Elastic lidar | Measurement in time of length of turbulent eddies |
| $\sigma_{RCS'}^2(z)$ | Elastic lidar | Homogeneity of aerosol distribution |
| $S_{RCS'}(z)$ | Elastic lidar | Aerosol motion (S < 0 ➜ Downdrafts, S> 0➜ Updrafts) |
| $K_{RCS'}(z)$ | Elastic lidar | Level of aerosol mixing (K < 3 ➜ Well-Mixed, K > 3 ➜ Low Mixing) |
| $PBLH_{Elastic}$ | Elastic lidar | Top of aerosol layer obtained from variance method |
| $PBLH_{MWR}$ | MWR | Top of CBL/SBL layer obtained from Potential Temperature |






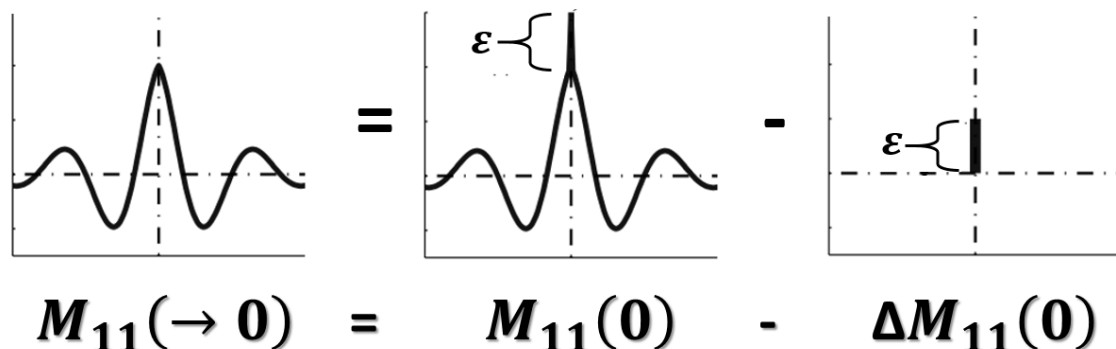

Figure 1 – Procedure to remove the errors of autocovariance functions. $M_{11}(\to 0)$ – corrected autocovariance function errors; $M_{11}(0)$ - autocovariance function without correction; $\Delta M_{11}(0)$ - error of autocovariance function.

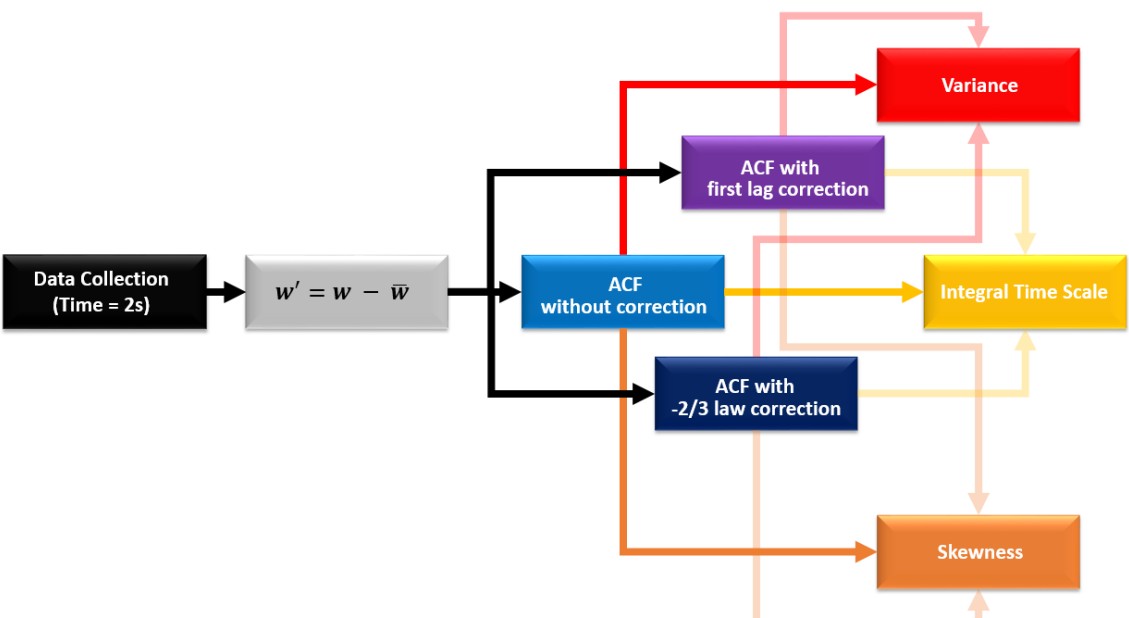

Figure 2 – Flowchart of data analysis methodology applied to the study of turbulence with Doppler lidar

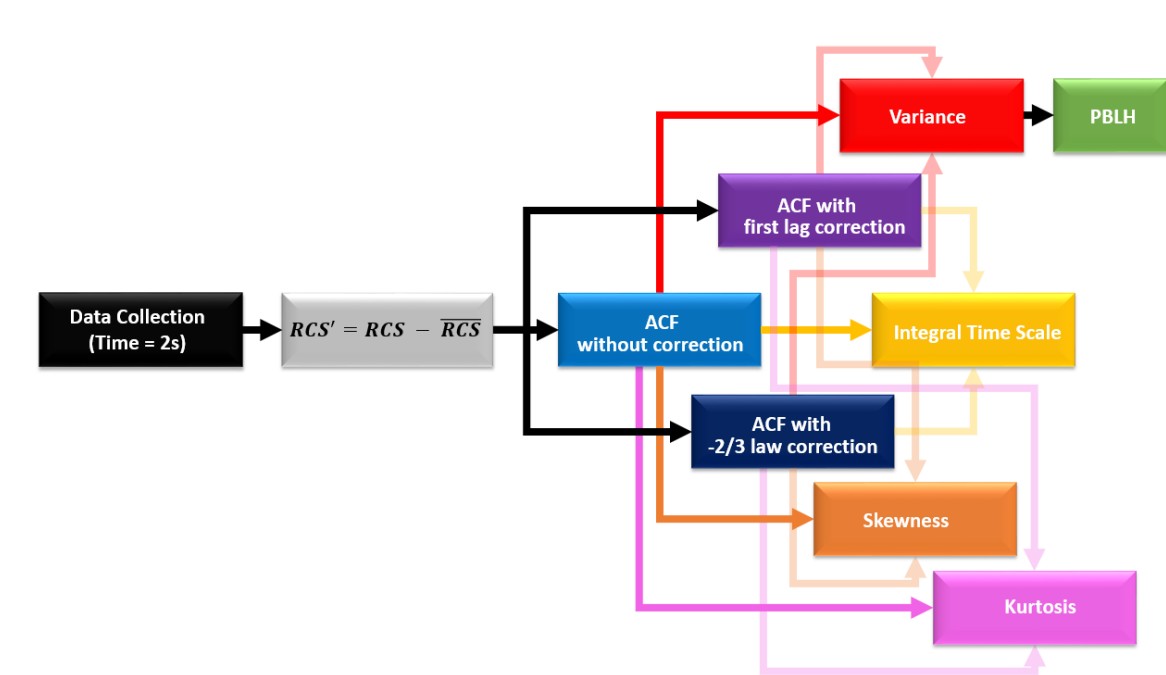

Figure 3 – Flowchart of data analysis methodology applied to the study of turbulence with elastic lidar

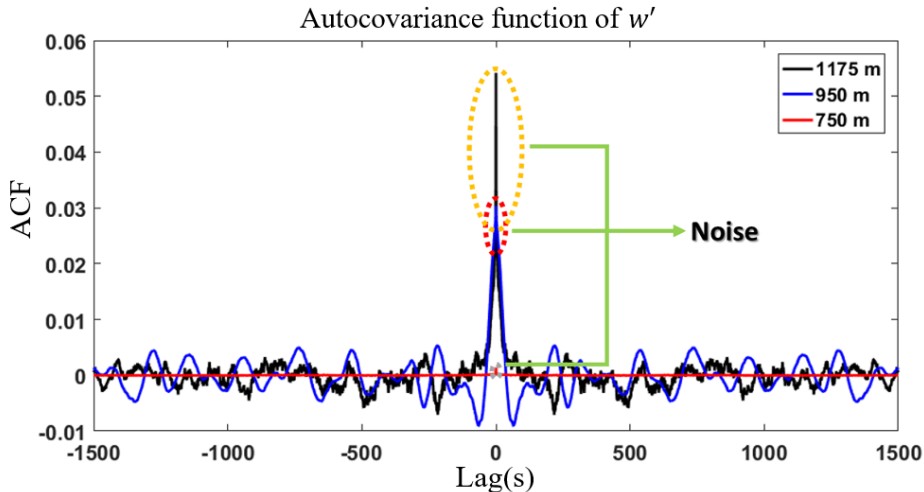

Figure 4 – Autocovariance function (ACF) of $w'$, obtained from Doppler lidar at three different heights on 19th May 2016 at 08-09 UTC in Granada.

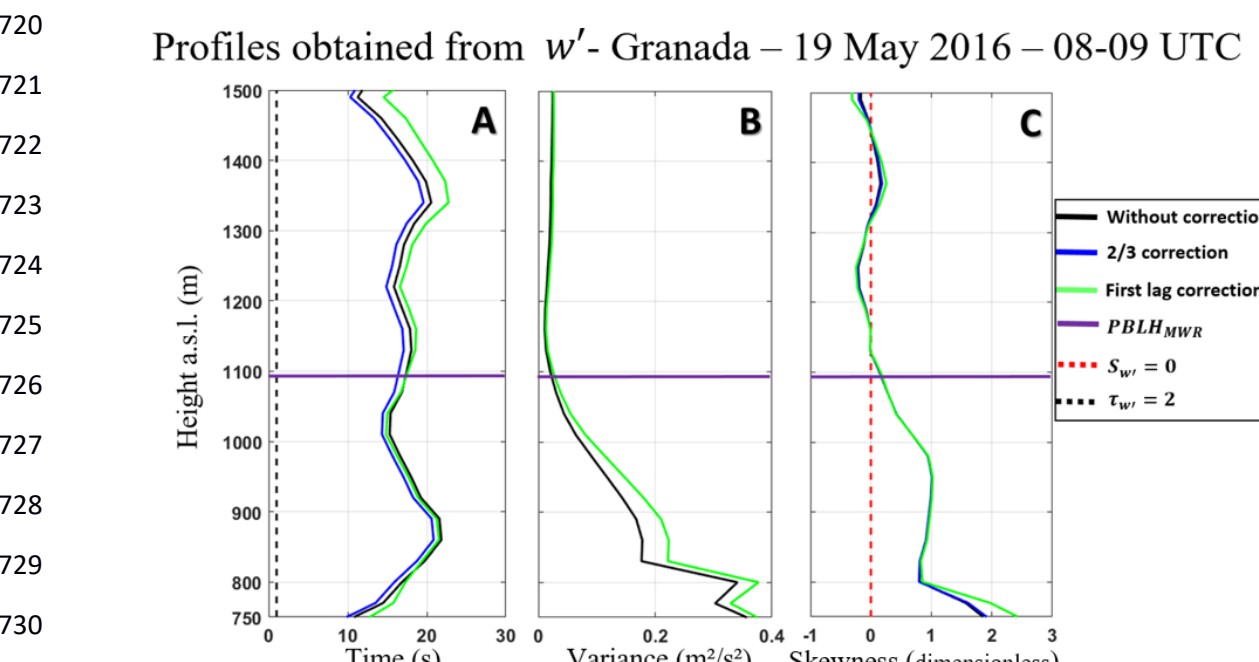

Figure 5 – A - Vertical profile of Integral time scale ($\tau_{w'}$). B - Vertical profile of variance ($\sigma^2_{w'}$). C - Vertical profile of Skewness ($S_{w'}$). All profiles were obtained from Doppler lidar data on 19th May 2016 at 08-09 UTC in Granada.

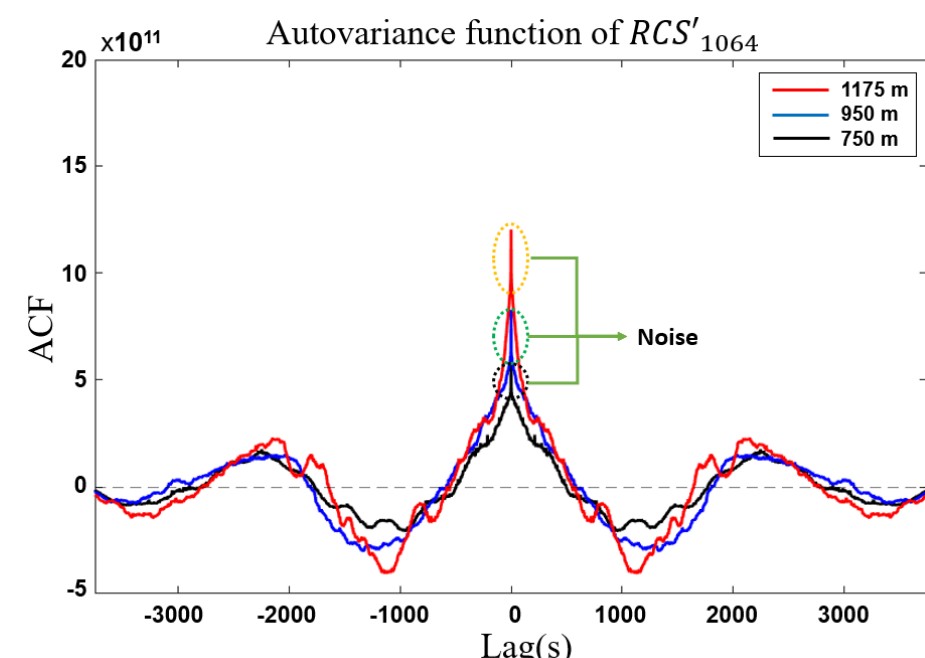

Figure 6 – Autocovariance of $RCS'_{1064}$ obtained from MULHACÉN elastic lidar data to three different heights on 19th May 2016 at 12-13 UTC in Granada.

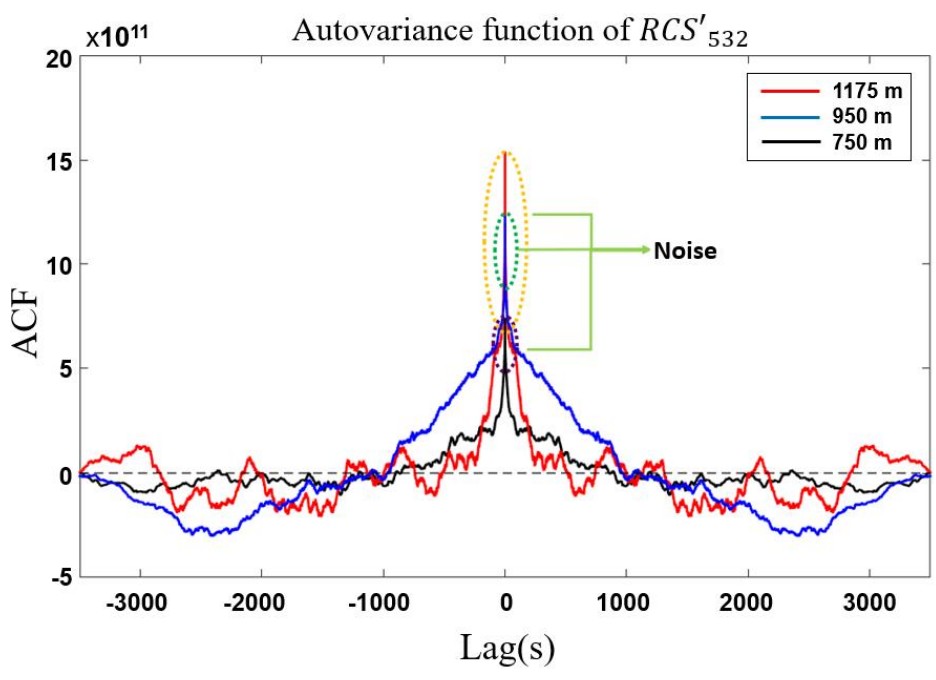

Figure 7 – Autocovariance of $RCS'_{532}$ obtained from MULHACÉN elastic lidar data to three different heights on 19th May 2016 at 12-13 UTC in Granada.

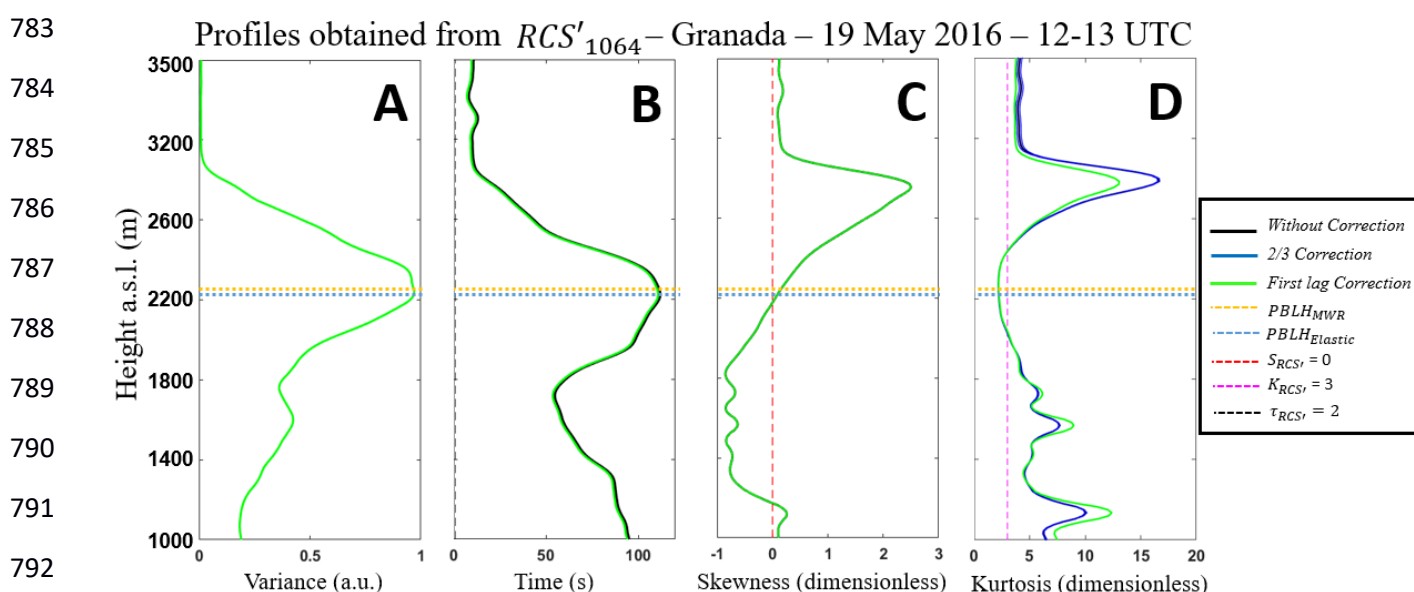

Figure 8 – A- Vertical profile of Integral time scale ($\tau_{RCS'}$). B - Vertical profile of variance ($\sigma^2_{RCS'}$). C - Vertical profile of Skewness ($S_{RCS'}$). D - Vertical profile of Kurtosis ($K_{RCS'}$). All profiles were obtained from MULHACÉN elastic lidar data on 19th May2016 in Granada between 12-13 UTC.

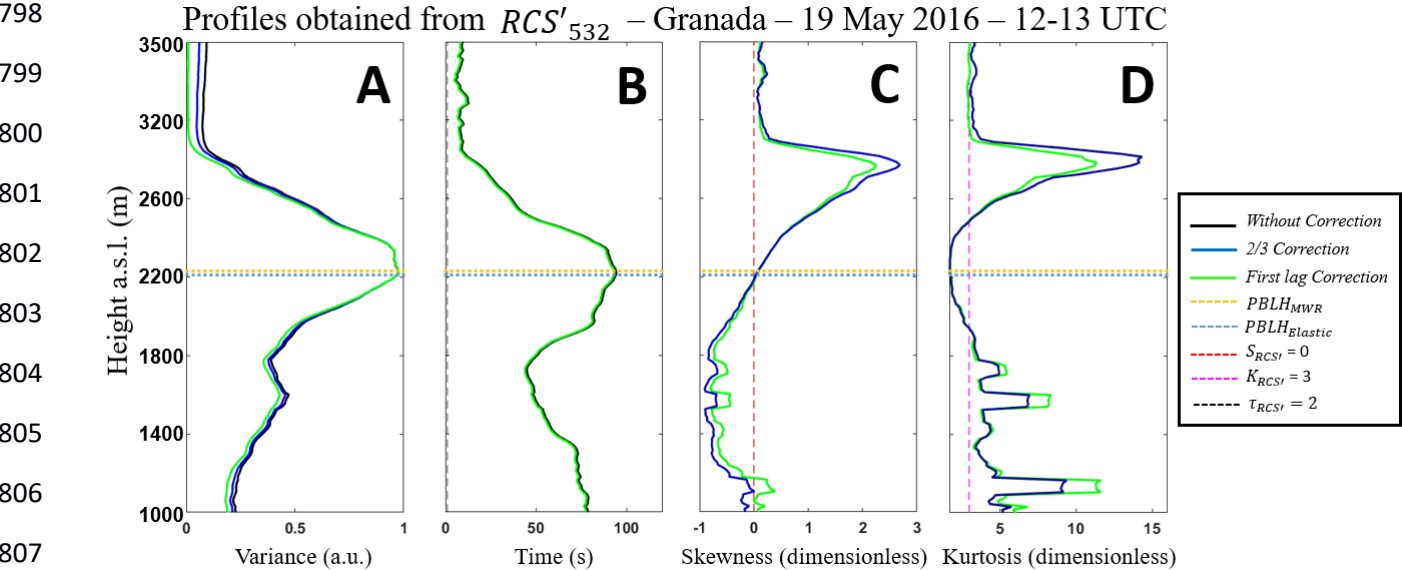

Figure 9 – A- Vertical profile of Integral time scale ($\tau_{RCS'}$). B - Vertical profile of variance ($\sigma^2_{RCS'}$). C - Vertical profile of Skewness ($S_{RCS'}$). D - Vertical profile of Kurtosis ($K_{RCS'}$). All profiles were obtained from MULHACÉN elastic lidar data on 19th May2016 in Granada between 12-13 UTC.

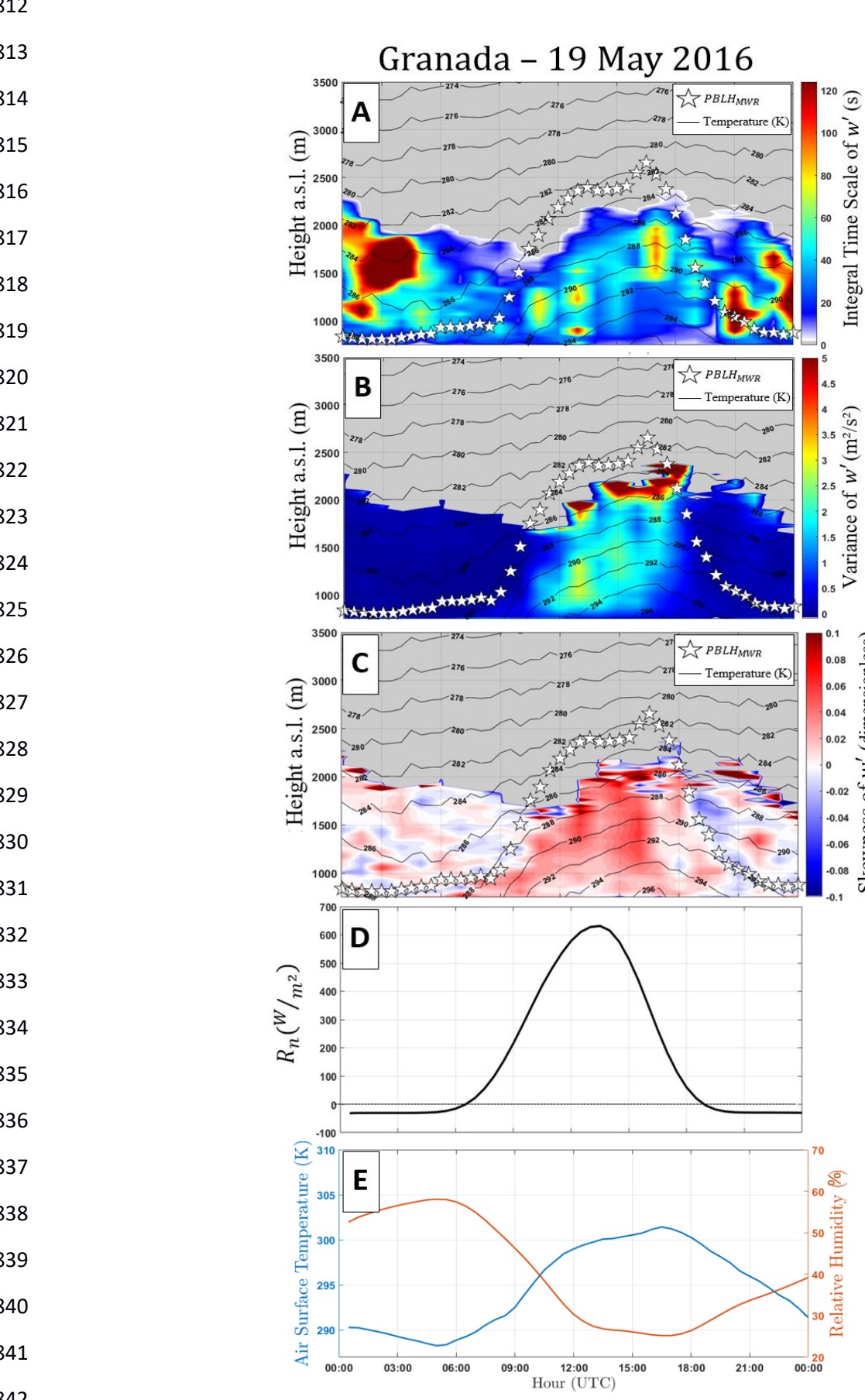

Figure 10 – A – integral time scale obtained from Doppler lidar data [$\tau_{w'}$], B – variance obtained from Doppler lidar data [$\sigma^2_{w'}$], C – skewness obtained from Doppler lidar data [$S_{w'}$], D – net radiation obtained from pyranometer data [$R_n$], E – Air surface temperature [blue line] and surface relative humidity [$RH$ - orange line] both were obtained from surface sensors. All profiles were acquired on 19th May 2016 in Granada. In A, B and C black lines and white stars represent air temperature and $PBLH_{MWR}$, respectively.

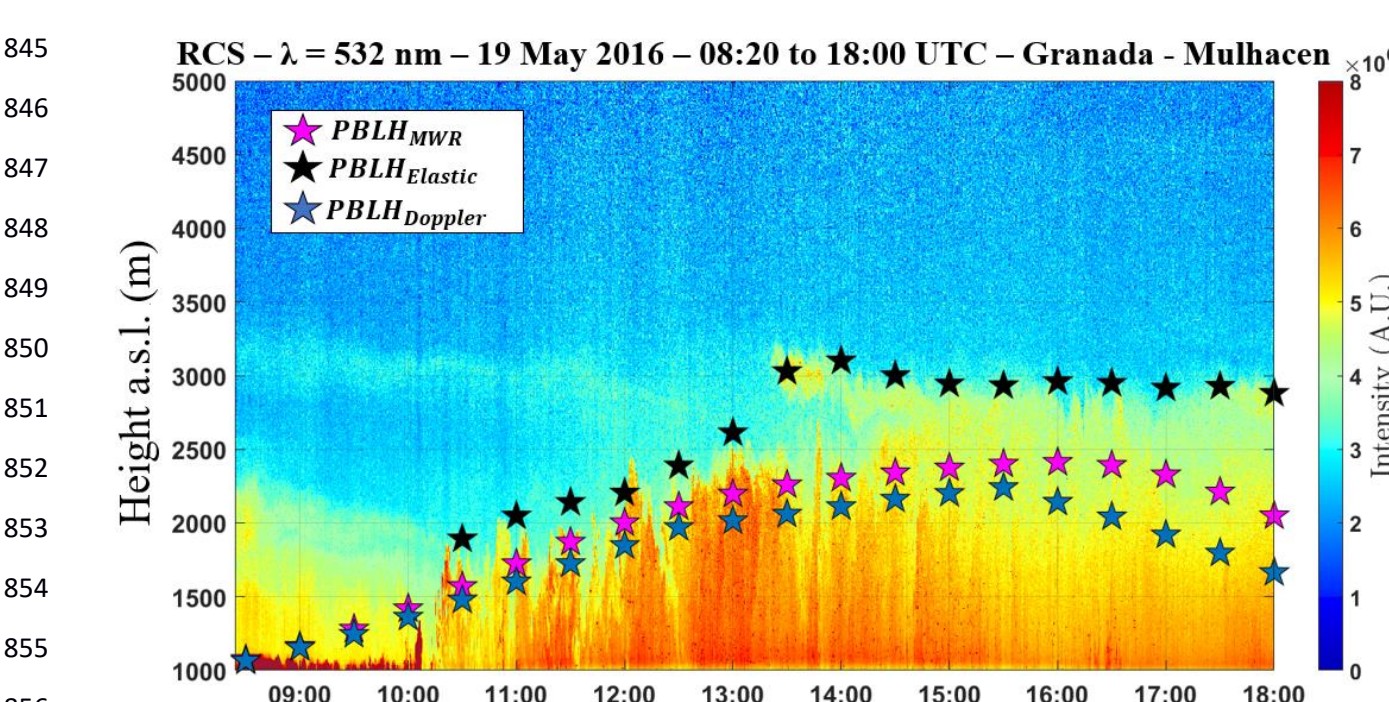

Figure 11 – Time-Height plot of RCS obtained on 19 May 2016 in Granada. Pink stars represent the $PBLH_{MWR}$, black stars represent the $PBLH_{Elastic}$ and blues stars represent the $PBLH_{Doppler}$.

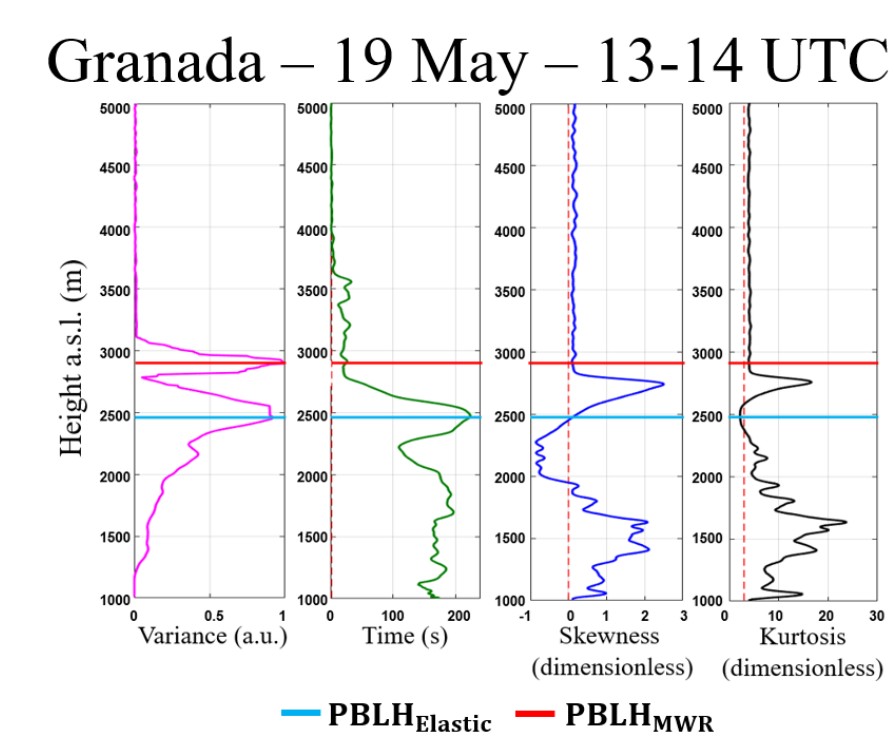

Figure 12 – Statistical moments obtained from 532 nm wavelength data of elastic lidar (MULHACÉN) in Granada at 13 to 14 UTC - 19 May 2016. From left to right: variance [$\sigma^2_{RCS'}$], integral time scale [$\tau_{RCS'}$], skewness [$S_{RCS'}$] and kurtosis [$K_{RCS'}$].

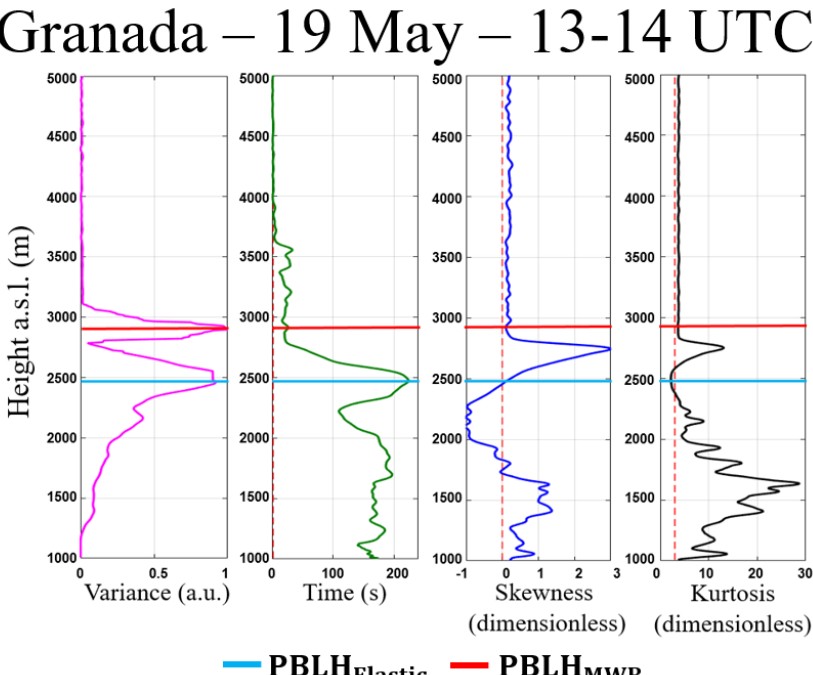

Figure 13 – Statistical moments obtained from 1064 nm wavelength data of elastic lidar(MULHACÉN) in Granada at 13 to 14 UTC - 19 May 2016. From left to right: variance [$\sigma^2_{RCS'}$], integral time scale [$\tau_{RCS'}$], skewness [$S_{RCS'}$] and kurtosis [$K_{RCS'}$].

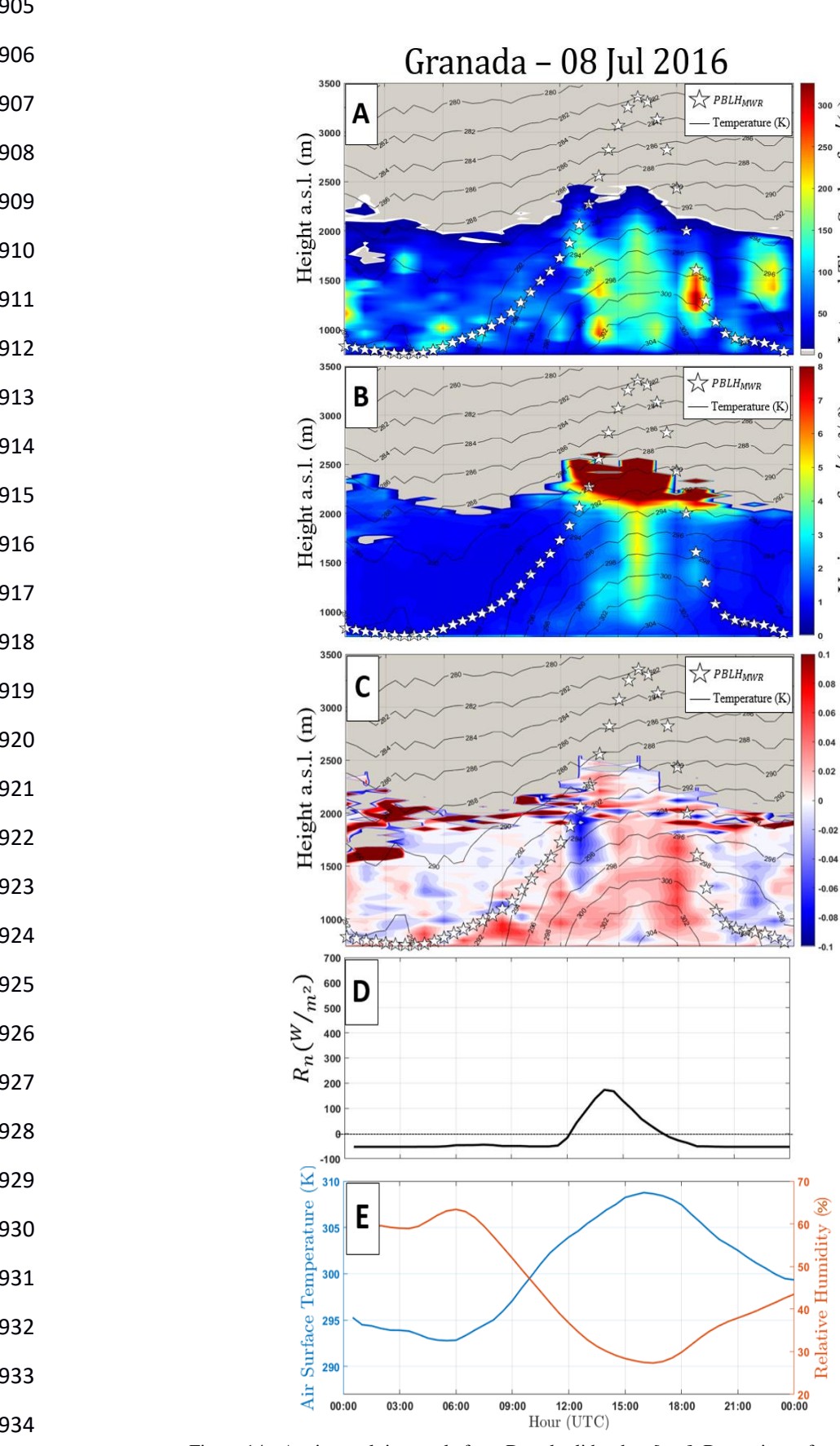

Figure 14 - A – integral time scale from Doppler lidar data [$\tau_{w'}$], B – variance from Doppler lidar data [$\sigma_{w'}^2$], C – skewness from Doppler lidar data [$S_{w'}$], D – net radiation from pyranometer data [$R_n$], E – Air surface temperature [blue line] and surface relative humidity [$RH$ – orange line] from surface sensor data. All profiles were obtained in Granada on 08 July 2016. In A, B and C black lines and white stars represent air temperature and $PBLH_{MWR}$, respectively.

936

937

938

939

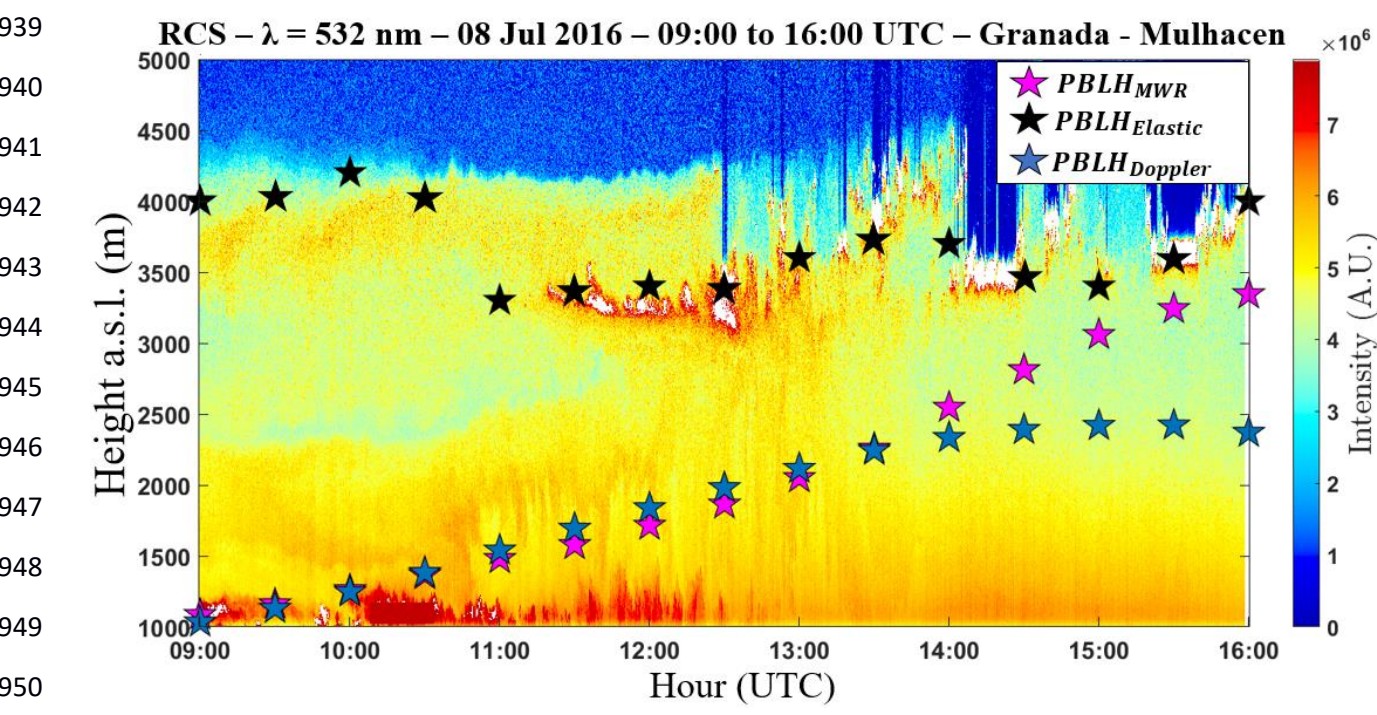

Figure 15 – Time-Height plot of RCS obtained from MULHACÉN elastic lidar data on 08 July 2016 in Granada. Pink stars represent the $PBLH_{MWR}$, black stars represent the $PBLH_{Elastic}$ and blues stars represent the $PBLH_{Doppler}$.

















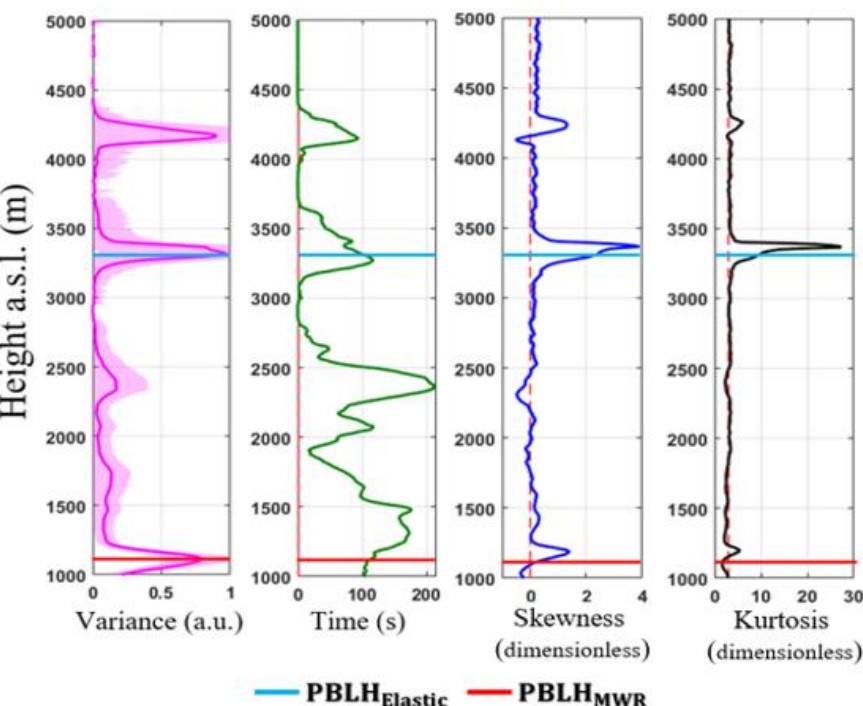

Figure 16 - Statistical moments obtained from 532 nm wavelength data of elastic lidar(MULHACÉN) in Granada between 11-12 UTC on 08th July 2016. From left to right: variance [$\sigma^2_{RCS'}$], integral time scale [$\tau_{RCS'}$], skewness [$S_{RCS'}$] and kurtosis [$K_{RCS'}$].

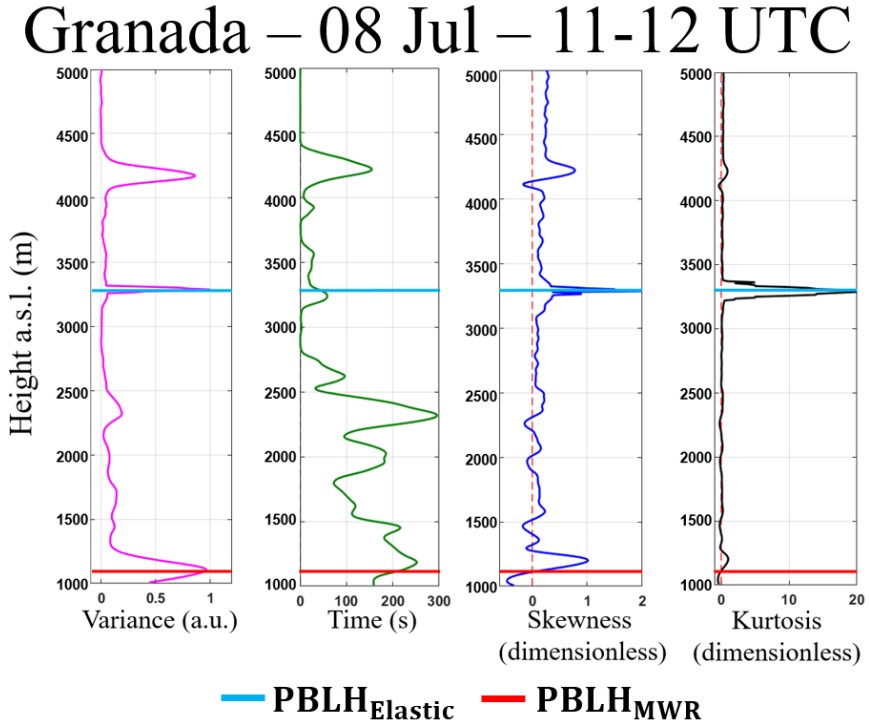

Figure 17 - Statistical moments obtained from 1064 nm wavelength data of elastic lidar(MULHACÉN) in Granada between 11-12 UTC on 08th July 2016. From left to right: variance [$\sigma^2_{RCS'}$], integral time scale [$\tau_{RCS'}$], skewness [$S_{RCS'}$] and kurtosis [$K_{RCS'}$].

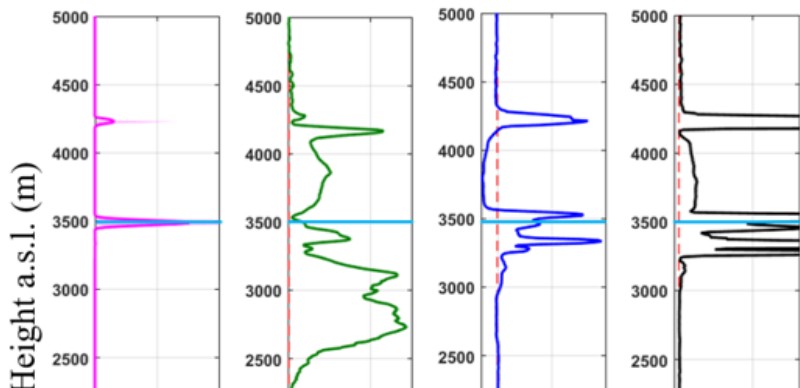

Figure 18 - Statistical moments obtained from 532 nm wavelength data of elastic lidar (MULHACÉN) in Granada between 12 -13 UTC on 08 July 2016. From left to right: variance [$\sigma^2_{RCS\prime}$], integral time scale [$\tau_{RCS\prime}$], skewness [$S_{RCS\prime}$] and kurtosis [$K_{RCS\prime}$].

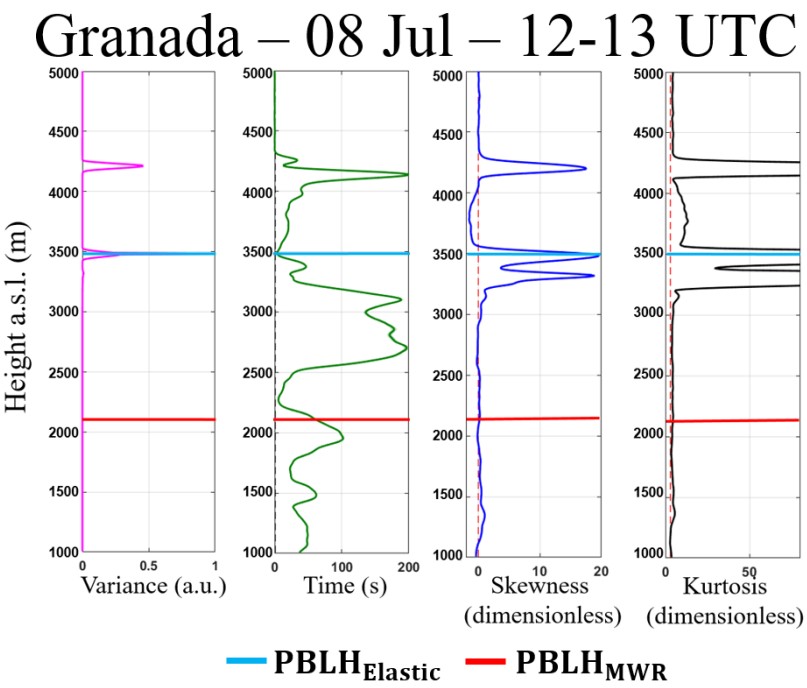

Figure 19 - Statistical moments obtained from 1064 nm wavelength data of elastic lidar (MULHACÉN) in Granada between 12 -13 UTC on 08 July 2016. From left to right: variance [$\sigma^2_{RCS\prime}$], integral time scale [$\tau_{RCS\prime}$], skewness [$S_{RCS\prime}$] and kurtosis [$K_{RCS\prime}$].