# Peer review of "Analyzing the turbulent Planetary Boundary Layer by"

_Atmospheric Chemistry and Physics, 2018_

## Referee Comment (RC1) · Anonymous Referee #2 · 25 May 2018

The paper presents different techniques for boundary layer detection with lidar and microwave radiometer. A method for error reduction in Doppler lidar and elastic lidar data is given. The paper is, however, not yet ready for publication. Several major issues must be met before publication:

Major issues:

1.) Different techniques for boundary layer detection are presented. This is however not new. An an actual comparison is not even presented: In all figures, the actual

boundary layer height is only shown for the microwave retrieval (stars). I would lilke to see at least one case study, in which all three techniques are shown together. Please put in symbols that indicate the boundary layer development for the Doppler lidar and the elastic lidar methods as well. So far, this information is only given in single profiles, which is not enough. The Doppler lidar also yields measurements of attenuated backscatter, so you actually can compare all three techniques in one case study.

2.) Three different definitions of boundary layer are used in the context of the paper: Thermodynamic (Temperature detected by the Microwave radiometer) Turbulence (Mixing Layer height detected by Doppler lidar) Aerosol load (Aerosol boundary layer and residual layers detected by elastic lidar) What use is it to compare measurements based on these three definitions? What does it mean, if the measurements are in agreement/disagreement?

3.) The message of the paper is not clear. Only the very last sentence refers to the larger meaning of the paper and indicates that something could be done in the context of the EARLINET network. Please elaborate more on why and how these techniques should be used together within EARLINET. What is the overarching goal? Which gaps are filled by the technique?

4.) The paper is of pure technical nature. This is usually not the scope of ACP, more of AMT or an ACP technical note. I leave it up to the associate editor to decide whether the paper may be suitable anyway for the EARLINET special issue.

Minor issues:

- Please update the flowcharts in Figs. 2 and 3 and give them a better layout: What does the blue circle mean? The symbols seem unnecessary. Boxes with text should be used instead. And please try to make the flowcharts in a way that there is a flow of information into one direction. Currently, the pathways are quite convoluted.

- L.281: "do not have significant differences": An actual time-height comparison of

PBLH is not given (see major point 1), so it is hard to assess what "significant" means in this context. There will be deviations, which should be quantified (height difference).

- L.286: "The skewness values obtaiend from RCS give us information about aerosol motion.": There is no way to detect aerosol motion directly with an elastic lidar. If there should be any statistical correlation between aerosol motion and skewness, please give references or discuss in the introduction. Moreira et al. (2018) is cited, but does not appear in the reference list. Please be more thorough, here. This topic is much too complicated to be handled with a single comment. Even if there should be any statistical relationship between backscatter skewness and turbulence it will depend on a lot of assumptions. And the discussion of those would make the paper even more blurry.

Technical corrections:

L. 19: aircrafts -> aircraft L. 25: compare -> compared L. 36: process -> processes L. 61: due to its -> due to their L. 62: "Several kind of tracers" -> I think this refers to the boundary layer height. Please think again about the term "tracers". In the context of this paper it could be misleading. E.g. for a Doppler lidar, aerosol particles are tracers for wind velocity. L. 89: Responsible of -> responsible for L. 94: "MULHACEN": Please describe the abbreviation L.103: Streamline -> If you want to name the instrument it would be nice to also name the sub-type (PRO, SR, XR, ...). The technical differences between the systems can be significant. L.130: PBLH is introduced without description L.189: LALINET: please describe this abbreviation L.229: Errors bars -> Error bars

---

## Referee Comment (RC2) · Anonymous Referee #1 · 10 Jul 2018

This manuscript presents results from the SLOPE campaign in Granada, Spain, in which the objective was to obtain closure between remote sensing and in-situ measurements. For this manuscript, the focus is on characterising the planetary boundary layer using a Doppler lidar, multi-wavelength lidar (MULHACEN), and a profiling microwave radiometer, all operating at high temporal resolution (2 seconds). The authors investigate the use of fluctuations in aerosol number density from the elastic system (EL), vertical velocity fluctuations obtained from the Doppler lidar (DL), and potential temperature profiles retrieved from the microwave radiometer (MWR), to identify the

boundary layer height (PBLH).

Some of the methodology is relevant, and the influence of random error introducing extra noise in higher-order moments is explored using suitable techniques, but the manuscript is not yet suitable for publication unless some major issues are addressed.

Major comments

The manuscript title and abstract suggests that different methods to determine PBLH will be combined synergistically, but this is not discussed at all in the main text. The main text seems to focus on whether various parameters derived from each instrument agree and does not suggest how they can be combined. In addition, the reader is not informed how PBLH should be derived from many of the DL and EL parameters, or how they could be combined if the purpose was to describe a synergistic retrieval method. Please decide whether you are describing a synergistic approach, or an intercomparison, and structure the manuscript accordingly.

The EL and DL parameters are calculated over 1-hour periods. Is this 1-hour timescale suitable during rapidly varying conditions such as during the morning growth of the boundary layer? Did you try using a running average? What is the impact if you change the averaging period, and why was 1-hour chosen when the MWR data are averaged over 30 minutes?

The manuscript requires a much more rigorous description of the processes driving turbulent mixing in the boundary layer. This does not need to be very long, but any processes referred to should be described accurately, e.g. it is the positive surface heat flux that is responsible for buoyancy (convection), not just intensifying convection. The energy flux balance at the surface partitions net radiation into sensible heat flux, latent heat flux and ground heat flux, hence, there can still be a positive sensible heat flux even when the net radiation is negative, such as during the early evening in urban regions, which is almost certainly what is happening in the two case studies shown here. It is not surprising that RH is somewhat inversely correlated with temperature

if the specific humidity mixing ratio remains constant; however it is not safe (and not necessary) to state anything about latent heat fluxes if you are not measuring them.

Minor comments

MWR data analysis: The MWR retrievals have, by some margin, the lowest vertical resolution of the methods detailed here, especially at the altitudes for typical daytime PBLH. The PBLH retrievals also seem very smooth in time. How does this compare with PBLH retrievals from DL and EL? Is it likely that the MWR provides the most accurate measure of PBLH? Do you use MWR PBLH as a reference for DL and EL retrievals or not? The manuscript requires some discussion on these issues.

Doppler lidar analysis: There are no time-height plots of the DL signal and velocity measurements so it is difficult to judge whether some of the features seen in the DL parameters are due to low SNR conditions. The interpretation of skewness is not appropriate and should be rewritten.

Elastic lidar analysis: Is it safe to assume the two-way transmittance is negligible? Especially since you use the 532 nm wavelength (molecular extinction may be important). What are the typical molecular, aerosol and total extinction values for the cases shown here? There are no time-height plots of the statistical parameters calculated from EL data so it is difficult to judge whether these provide a reliable guide to the boundary layer development - please include these.

Doppler lidar and Elastic lidar analysis: Since you make some effort to quantify the influence of noise on the statistical parameters derived from these two systems, it would be beneficial to discuss how this impacts your interpretation, e.g include time-height plots of the correction factor or relative correction, relative importance in determining PBLH, how much temporal averaging is required to obtain good results. What is the minimum integral time scale that the DL and EL can measure? Is it the acquisition time that allows you to observe turbulence throught the PBL, or is it more likely to be a function of the instrument sensitivities?

Case study 2: Did you try cloud-screening EL data before calculating EL parameters? The PBLH from EL would agree much better with PBLH from MWR in Figure 13, and maybe Figure 14 (it is hard to tell with the scales used). Clouds should also be visible in DL data.

Technical comments Line 36: What do you mean by cyclic processes?

Line 37: Large variability of what?

Line 39: Surface heating is unlikely to impact the upper troposphere.

Line 84: Distinct?

Line 89: Replace 'responsible of' with 'responsible for'.

Line 98: Explain '(s and p)'.

Line 104: Please include a few more Doppler lidar operating parameters: pulse repetition frequency, telescope focus.

Line 106: Use 'laser beam pointing at vertical', since the ground surface may not be horizontal!

Line 108: Replace 'which is part of the MWRNet' with 'which is a member of MWRNet'.

Line 112: State how many frequencies measured in each band.

Line 128: Replace 'MWR data analyzes' with 'MWR data analysis'

Line 130: PBLH not defined yet.

Line 188: Do you mean '(Pal et al., 2010)'?

Line 220: Replace 'Under' with 'Below'.

Lines 220-221: This sentence does not make sense. Do you mean 'Below the PBLH_MWR, correcting for noise does not have a significant impact on the profile, but is more evident above'?

Line 261: Define Rn (presumably net surface radiation).

Line 320: Do you mean '(Ansmann, 2010)'?

Figure 4: Autocovariance from DL? What are the units for variance and skewness?

Figures 5,7: Profiles from which instrument, and from which location? At what time, and on what day? What height is the surface?

Figure 6: Autocovariance from EL? What are the units for variance, skewness and kurtosis?

Figures 8,11: Which instrument are panels A-C from? Are the black lines (temperature) from the MWR retrieval? Is it more appropriate to plot variance in log scale?

Figure 9,12: Which instrument is this figure from? This is a time-height plot of RCS, not a profile.
* * *

---

## Author Comment (AC1) · 14 Aug 2018

**Analyzing the turbulence in the Planetary Boundary Layer by the synergic use of remote sensing systems: Doppler wind lidar and aerosol elastic lidar" *by* Gregori de Arruda Moreira et al.**

**Author's response**

We thank the anonymous reviewers for their comments, corrections and suggestions, which have helped to improve the quality of the manuscript. According to the referees' reports, the following changes have been performed on the original manuscript and a point-by-point response is included below.

**Anonymous Referee #2**

The paper presents different techniques for boundary layer detection with lidar and microwave radiometer. A method for error reduction in Doppler lidar and elastic lidar data is given. The paper is, however, not yet ready for publication. Several major issues must be met before publication:

**Major issues:**

1.) Different techniques for boundary layer detection are presented. This is however not new. An actual comparison is not even presented: In all figures, the actual boundary layer height is only shown for the microwave retrieval (stars). I would like to see at least one case study, in which all three techniques are shown together. Please put in symbols that indicate the boundary layer development for the Doppler lidar and the elastic lidar methods as well. So far, this information is only given in single profiles, which is not enough. The Doppler lidar also yields measurements of attenuated backscatter, so you actually can compare all three techniques in one case study.

We thank the Reviewer 2 for this comment. However, in this paper we do not present different techniques for planetary boundary layer height detection. That issue have been covered in a paper recently published by some of the authors of this study (Moreira et al., 2018a). Our focus is not comparing different retrieval schemes but approaching the synergetic combination of information from different remote sensing systems that are sensitive to different tracers, in order to describe some turbulent process that occur during the PBL daily cycle. This information is also used for discussing on the influence of different factors on the PBL behavior. In order to clarify this point the following changes have been implemented:

- New Title: *"Analyzing the turbulent Planetary Boundary Layer behavior by the synergic use of remote sensing systems: Doppler wind lidar, aerosol elastic lidar and microwave radiometer"*
- Abstract:
(Page 1, line 23)

*"In this study we propose the synergetic use of remote sensing systems (microwave radiometer [MWR], Doppler lidar [DL] and elastic lidar [EL]) to analyze the turbulent PBL behavior. Furthermore, we show how some meteorological variables such as air temperature, aerosol number density, vertical wind speed, relative humidity and net radiation might influence the turbulent PBL dynamic."*

(Page 1, line 28)

*"In both cases the results provided by the different instruments are complementary. Thus, the synergistic use of the different systems allow us performing a detailed monitoring of the turbulent PBL behavior, as well as a better understanding about how the analyzed variables can interfere in this process."*

(Page 11, line 376)

*"In this paper we analyze the turbulent PBL behavior and how each detected variable can influence it. Such observations were made from the synergy of three different types of remote sensing systems (DL, EL and MWR) and surface sensors during SLOPE-I campaign."*

2.) Three different definitions of boundary layer are used in the context of the paper: Thermodynamic (Temperature detected by the Microwave radiometer) Turbulence (Mixing Layer height detected by Doppler lidar) Aerosol load (Aerosol boundary layer and residual layers detected by elastic lidar) What use is it to compare measurements based on these three definitions? What does it mean, if the measurements are in agreement/disagreement?

Our intention in this work is to perform a synergistic study of the PBL behavior based on different variables and, thus, based on different tracers. Therefore, the information obtained is complementary and allows for a detailed description about the turbulent PBL dynamics.

Therefore, we refer to "good agreement" when different variables (obtained from different instruments) provide results that complement each other. For example, the rise of CBL detected by MWR together with positive values of Skewness of w' and positive values of RCS' allows to conclude that CBL growth occurs, which is directly associated with convective process and causing the rising of aerosol plumes. Therefore, as these three results complement each other, we say that there is a good agreement, since they allow us to observe with a better panorama the turbulent processes in the PBL.

3.) The message of the paper is not clear. Only the very last sentence refers to the larger meaning of the paper and indicates that something could be done in the context of the EARLINET network. Please elaborate more on why and how these techniques should be used together within EARLINET. What is the overarching goal? Which gaps are filled by the technique?

We thank the Reviewer 2 for this comment. In order to clarify this point the following changes have been implemented:

(Page 2, line 73)

*"However, this subject requires more exploration, mainly the synergy among lidar and others remote sensing systems, like microwave radiometer. Thus, the combination of information obtained from these instruments can provide a more detailed understanding about the turbulent PBL behavior. Such approach is even more attractive when considering facilities of networks, e. g. European Aerosol Research Lidar NETwork (EARLINET) (Pappalardo et al., 2014), Microwave Radiometer Network (MWRNET) (Rose et al., 2005; Caumont et al., 2016) and ACTRIS CLOUDNET (Illingworth et al., 2007)."*

(Page 5, line 221)

*"These three methodologies, together with data of net surface radiation (obtained from pyranometer data) and air temperature (provided by MWR), are used synergistically in order to complement one each other and consequently generate a detailed picture of how each variable influences the turbulent PBL behavior, as it will be demonstrated in subsection 4.2."*

4.) The paper is of pure technical nature. This is usually not the scope of ACP, more of AMT or an ACP technical note. I leave it up to the associate editor to decide whether the paper may be suitable anyway for the EARLINET special issue.

We thank the Reviewer 2 for this comment. We have submitted this manuscript to the EARLINET special issue in order to show how EARLINET can contribute to provide useful information on the PBL structure using the available instrumentation in combination with new measurement protocols and combining lidar information with that retrieved with complementary remote sensing systems, included in some EARLINET stations and particularly in those that are ACTRIS CLOUDNET stations (Illingworth et al., 2007).

*Illingworth, A. J., Hogan, R. J. O' Connor, E. J. Bouniol, D. Brooks, M. E. Delanoe, J. Donovan, D. P. Eastment, J. D. Gaussiat, N. Goddard, J. W. F. Haeffelin, M. Klein Baltink, H. Krasnov, O. A. Pelon, J. Piriou, J.-M. Protat, A. Russchenberg, H. W. J. Seifert, A. Tompkins, A. M. Van Zadelhoff, G.-J. Vinit, F. Willen, U. Wilson, D. R. and Wrench, C . L.: CLOUDNET: Continuous Evaluation of Cloud Profiles in Seven Operational Models using Ground-Based Observations.*

*Bulletin of the American Meteorological Society, 88, 883-898, doi:10.1175/BAMS-88-6-883, 2007.*

**Minor issues:**

- Please update the flowcharts in Figs. 2 and 3 and give them a better layout: What does the blue circle mean? The symbols seem unnecessary. Boxes with text should be used instead. And please try to make the flowcharts in a way that there is a flow of information into one direction. Currently, the pathways are quite convoluted.

We thank the Reviewer 2 for the comments. In order to clarify this point, the figure 2 and 3 have been changed as follows:

[Figure]

Figure 2 – Flowchart of data analysis methodology applied to the study of turbulence with Doppler lidar

[Figure]

Figure 3 – Flowchart of data analysis methodology applied to the study of turbulence with elastic lidar

- L.281: "do not have significant differences": An actual time-height comparison of PBLH is not given (see major point 1), so it is hard to assess what "significant" means in this context. There will be deviations, which should be quantified (height difference).

We thank the Reviewer 2 for the comments. In order to clarify this point, the text has been changed as follows:

(Page 9, line 284)

*"The maximum for the variance of RCS can be used as indicator of PBLH ($PBLH_{Elastic}$) (Moreira et al., 2015). Thus, the red line in all graphics represent the $PBLH_{Elastic}$ (2200 m a.s.l.) and the blue one the average value of $PBLH_{MWR}$ (2250 m a.s.l.), both obtained between 13 and 14 UTC.*

*Due to well-defined PBL, $PBLH_{Elastic}$ and $PBLH_{MWR}$ do not have significant differences (around 50 m)."*

- L.286: "The skewness values obtained from RCS give us information about aerosol motion.": There is no way to detect aerosol motion directly with an elastic lidar. If there should be any statistical correlation between aerosol motion and skewness, please give references or discuss in the introduction. Moreira et al. (2018) is cited, but does not appear in the reference list. Please be more thorough, here. This topic is much too complicated to be handled with a single comment. Even if there should be any statistical relationship between backscatter skewness and turbulence it will depend on a lot of assumptions. And the discussion of those would make the paper even more blurry.

We thank the Reviewer 2 for the comments. In order to clarify this point, the text has been changed as follows:

(Page 2, line 68)

*"For example, Pal et al. (2010) demonstrated how the statistical analyses obtained from high-order moments of elastic lidar can provide information about aerosol plume dynamics in the PBL region. In addition, when different lidar systems operate synergistically, as for example in Engelmann et al. (2008), who combined elastic and Doppler lidar data, it is possible to identify very complex variables such as vertical particle flux."*

(Page 14, line 452)

*Moreira, G. de A., Lopes, F. J. S., Guerrero-Rascado, J. L., Landulfo, E., Alados-Arboledas, L. Analyzing turbulence in Planetary Boundary Layer from multiwavelenght lidar system: impact of wavelength choice. Optics Express, submitted, 2018b.*

**Technical corrections:**

L.19: aircrafts -> aircraft

Done.

L.25: compare -> compared

Done.

L.36: process -> processes

Done.

L.61: due to its -> due to their

Done.

L.62: "Several kind of tracers" -> I think this refers to the boundary layer height. Please think again about the term "tracers". In the context of this paper it could be misleading. E.g. for a Doppler lidar, aerosol particles are tracers for wind velocity.

We thank the Reviewer 2 for the comments. In order to clarify this point, the text has been changed as follows:

*(Page 2, Line 62)*

*"...large vertical range, high data acquisition rate and capability to detect several observed quantities such as vertical wind velocity..."*

L.89: Responsible of -> responsible for

Done.

L.94: "MULHACEN": Please describe the abbreviation

We thank the Reviewer 2 for the comments. MULHACEN is the name of a mountain, which is part of the Sierra Nevada. This name was used to baptize our Multiwavelength Raman lidar, not corresponding to an acronym.

L.103: Streamline -> If you want to name the instrument it would be nice to also name the sub-type (PRO, SR, XR, ...). The technical differences between the systems can be significant.

We thank the Reviewer 2 for the comment. In order to clarify this point, the text has been changed as follows:

*(Page 2, Line 104)*

*"...model Stream Line XR..."*

L.130: PBLH is introduced without description

We thank the Reviewer 2 for the comments. In order to clarify this point, the text has been changed as follows:

*(Page 4, Line 131)*

*"...PBL Height (PBLH$_{MWR}$)..."*

L.189: LALINET: please describe this abbreviation

We thank the Reviewer 2 for the comments. In order to clarify this point, the text has been changed as follows:

*(Page 6, Line 200)*

*"...LALINET (Latin American LIdar NETwork)..."*

L.229: Errors bars -> Error bars

Done.

[revised manuscript text omitted]

---

## Author Comment (AC2) · 14 Aug 2018

**Analyzing the turbulence in the Planetary Boundary Layer by the synergic use of remote sensing systems: Doppler wind lidar and aerosol elastic lidar" *by* Gregori de Arruda Moreira et al.**

**Author's response**

We thank the anonymous reviewers for their comments, corrections and suggestions, which have helped to improve the quality of the manuscript. According to the referees' reports, the following changes have been performed on the original manuscript and a point-by-point response is included below.

**Anonymous Referee #1**

This manuscript presents results from the SLOPE campaign in Granada, Spain, in which the objective was to obtain closure between remote sensing and in-situ measurements. For this manuscript, the focus is on characterizing the planetary boundary layer using a Doppler lidar, multi-wavelength lidar (MULHACEN), and a profiling microwave radiometer, all operating at high temporal resolution (2 seconds). The authors investigate the use of fluctuations in aerosol number density from the elastic system (EL), vertical velocity fluctuations obtained from the Doppler lidar (DL), and potential temperature profiles retrieved from the microwave radiometer (MWR), to identify the boundary layer height (PBLH).

Some of the methodology is relevant, and the influence of random error introducing extra noise in higher-order moments is explored using suitable techniques, but the manuscript is not yet suitable for publication unless some major issues are addressed.

**Major comments**

The manuscript title and abstract suggests that different methods to determine PBLH will be combined synergistically, but this is not discussed at all in the main text. The main text seems to focus on whether various parameters derived from each instrument agree and does not suggest how they can be combined. In addition, the reader is not informed how PBLH should be derived from many of the DL and EL parameters, or how they could be combined if the purpose was to describe a synergistic retrieval method. Please decide whether you are describing a synergistic approach, or an intercomparison, and structure the manuscript accordingly.

We thank the Reviewer 1 for these comments. Such comments are in agreement with Referee#2, and have been accordingly addressed in section "Anonymous reviewer#2 - Major issue 1". Thus, we would like to remark that the objective of this work is not to compare the PBLH retrievals from different instrument because it has been demonstrated

in previous articles (e.g. Bravo-Aranda et al, 2017; Moreira et al., 2018a) that instruments and techniques based on different tracers and observed quantities retrieve the height of different PBL sublayers. In this way, we propose here to use in a synergic way the information on the PBL obtained by these different remote sensing techniques in order to get a better understanding on the evolution of the PBL.

The EL and DL parameters are calculated over 1-hour periods. Is this 1-hour timescale suitable during rapidly varying conditions such as during the morning growth of the boundary layer? Did you try using a running average? What is the impact if you change the averaging period, and why was 1-hour chosen when the MWR data are averaged over 30 minutes?

We thank the Reviewer 1 for this comment. We performed some tests with smaller time scales, such as 30 and 45 minutes. However, the influence of noise is larger and the obtained values of the integral time scale are lower. As we argued previously, here we do not do a comparison among MWR and the other remote sensing systems retrievals and, therefore, the different time scale does not interfere in our analysis.

The manuscript requires a much more rigorous description of the processes driving turbulent mixing in the boundary layer. This does not need to be very long, but any processes referred to should be described accurately, e.g. it is the positive surface heat flux that is responsible for buoyancy (convection), not just intensifying convection. The energy flux balance at the surface partitions net radiation into sensible heat flux, latent heat flux and ground heat flux, hence, there can still be a positive sensible heat flux even when the net radiation is negative, such as during the early evening in urban regions, which is almost certainly what is happening in the two case studies shown here. It is not surprising that RH is somewhat inversely correlated with temperature if the specific humidity mixing ratio remains constant; however it is not safe (and not necessary) to state anything about latent heat fluxes if you are not measuring them.

We thank the Reviewer 1 for the comments. In order to clarify this point, the text has been changed as follows:

*(Page 8, Line 272)*

*"This process is in agreement with the behavior of skewness of $w'$ ($S_{w'}$) shown in Figure 8-C. $S_{w'}$ is directly associated with the direction of turbulent movements. Thus, positive values correspond with a surface-heating-driven boundary layer, while negative ones are associated to cloud-top long-wave radiative cooling. If $S_{w'}$ is positive, both $\sigma_{w'}^2$ and TKE (Turbulent Kinetic Energy) are being transported upwards and, consequently, the red regions in Figure 13-C represent positive values of $S_{w'}$ and the blue regions refer to negative ones. During the stable period, there is predominance of low values of $S_{w'}$. Nevertheless, as air temperature increases (transition from stable to unstable period), $S_{w'}$ values begin to become positive and increase with the ascent of the $PBLH_{MWR}$ (CBL). Air temperature begins to decrease around 18:00 UTC, causing the reduction of $S_{w'}$. In this moment the transition from unstable to stable period occurs and, therefore, the reduction in $PBLH_{MWR}$ is due to the SBLH detection.*

*Figure 8-D shows the values of net surface radiation ($R_n$) that are estimated from solar global irradiance values using the seasonal model described in Alados et al. (2003). The negative values of $R_n$ are concentrated in the stable region. $R_n$ begins to increase around 06:00 UTC and reaches its maximum in the middle of the day. Comparing figures 8-C and 8-D, we can observe similarity among the behavior of $S_{w'}$, $R_n$ and surface air temperature, because these variables increase and decrease together, as expected.*

*The increase of $R_n$ causes the rise of surface air temperature, which contributes to the positive latent heat flux from the surface ($S_{w'}$) and, consequently, the growth of the $PBLH_{MWR}$ (CBL). $R_n$ begins to decrease certain time before the other variables, but the intense reduction of air temperature and decrease of $S_{w'}$ and SBLH detection occurs when $R_n$ becomes negative again, although there can still be a positive sensible heat flux, what is characteristic of early evening in urban regions due to the release of the ground heat flux at that time.*

*Figure 8-E presents the values of surface air temperature and surface relative humidity (RH). Air surface temperature is directly related with $R_n$ and $S_{w'}$ values, as afore mentioned and expected. On the other hand, RH is inversely correlated with temperature and, thus, with the rest of variables, due to the relative constancy of the water vapor mixing ratio characteristic of our site during the study."*

**Minor comments**

MWR data analysis: The MWR retrievals have, by some margin, the lowest vertical resolution of the methods detailed here, especially at the altitudes for typical daytime PBLH. The PBLH retrievals also seem very smooth in time. How does this compare with PBLH retrievals from DL and EL? Is it likely that the MWR provides the most accurate measure of PBLH? Do you use MWR PBLH as a reference for DL and EL retrievals or not? The manuscript requires some discussion on these issues.

We thank the Reviewer 1 for this comment. As aforementioned, in this paper we do not aim to perform a comparison of PBLH obtained from different method or instruments. This is something that we did in a previous paper: de Arruda Moreira et al. (2018a).

We use the MWR data as reference to estimate the PBLH due to a comparison between MWR and radiosonde data performed during a 3-month campaign in Granada-Spain (Moreira et al., 2018a). This comparison demonstrated a good correlation between these instruments in stable and convective situations.

In order to clarify this point, the text has been changed as follows:

(Page 5, line 152)

*"This methodology of PBLH detection was selected as the reference due to the results obtained during a performed campaign of comparison between MWR and radiosonde data, where twenty-three radiosondes were launched. High correlations were found between PBLH retrievals provided by both instruments in stable and unstable cases. Further details are given by Moreira et al. (2018a)."*

Doppler lidar analysis: There are no time-height plots of the DL signal and velocity measurements so it is difficult to judge whether some of the features seen in the DL parameters are due to low SNR conditions. The interpretation of skewness is not appropriate and should be rewritten.

We thank the Reviewer 1 for the comments. In order to clarify this point, the text has been changed as follows:

*(Page 8, Line 272)*

*"This process is in agreement with the behavior of skewness of $w'$ ($S_{w'}$) shown in Figure 8-C. $S_{w'}$ is directly associated with the direction of turbulent movements. Thus, positive values correspond with a surface-heating-driven boundary layer while negative ones are associated to cloud-top long-wave radiative cooling. If $S_{w'}$ is positive, both $\sigma^2_{w'}$ and TKE (Turbulent Kinetic Energy) are being transported upwards and consequently, the red regions in Figure 13-C represent positive values of $S_{w'}$ and the blue regions refer to negative ones."*

In addition, the following figures will be provided as supplementary materials:

[Figure]

Supplementary Material – Figure 1

[Figure]

Supplementary Material – Figure 2

[Figure]

Supplementary Material – Figure 3

[Figure]

Supplementary Material – Figure 4

Elastic lidar analysis: Is it safe to assume the two-way transmittance is negligible? Especially since you use the 532 nm wavelength (molecular extinction may be important). What are the typical molecular, aerosol and total extinction values for the cases shown here?

Yes. In order to clarify this point, the text has been changed as follows:

*(Page 9, Line 300)*

*"The period between 13:00 and 14:00 UTC has been selected to be analyzed. Figure 10-A presents the profiles of molecular ($\beta_{Molecular}$) and aerosol ($\beta_{Aerosol}$) backscatter coefficients at 532 nm. Although $\beta_{532}$ is composed by $\beta_{Molecular}$ and $\beta_{Aerosol}$, it is possible to observe the predominance of $\beta_{Aerosol}$ in the region below of the $PBLH_{MWR}$, as demonstrated in figure 10-B by the $\beta_{Ratio}$ profile . Similar results were demonstrated by Moreira et al. (2018b), therefore reinforcing the viability of the use of this wavelength in studies about turbulence."*

**Granada – 19 May – 13-14 UTC**

Figure 10 – (A) $\beta_{Molecular}$ (blue line) and $\beta_{Aerosol}$ (orange line). (B) $\beta_{Ratio}$ (black line). All profiles were obtained from the 532 nm lidar signal

*(Page 11, Line 357)*

*"Figure 14-A presents the $\beta_{Molecular}$ and $\beta_{Aerosol}$ profiles, similarly to figure 10-A. It is evident the predominance of $\beta_{Aerosol}$ in the region below $PBLH_{MWR}$, as demonstrated by $\beta_{Ratio}$ profile in figure 14-B. However due to presence of dust layer this dominance of $\beta_{Aerosol}$ is extended to approximately 4500 m a.s.l. Therefore the methodology proposed by Moreira et al. (2018b), based on considerations of Pal et al. (2010), can be applied."*

[Figure]

Figure 14 – (A) $\beta_{Molecular}$ (blue line) and $\beta_{Aerosol}$ (orange line). (B) $\beta_{Ratio}$ (black line). All profiles were obtained from the 532 nm lidar signal

There are no time-height plots of the statistical parameters calculated from EL data so it is difficult to judge whether these provide a reliable guide to the boundary layer development - please include these.

We thank the Reviewer 1 for the comments. In order to clarify this point, we have been added two figures in supplementary material, which demonstrate that all profiles maintain theirs specific behavior, as commented in page 9 – line 310, during the PBLH evolution.

[Figure]

Supplementary Material - Figure 5 - Statistical moments obtained from elastic lidar data at 14 to 15 UTC - 19 May 2016. From left to right: variance [$\sigma^2_{RCS'}$], integral time scale [$\tau_{RCS'}$], skewness [$S_{RCS'}$] and kurtosis [$K_{RCS'}$].

[Figure]

Supplementary Material - Figure 6 - Statistical moments obtained from elastic lidar data at 15 to 16 UTC - 19 May 2016. From left to right: variance [$\sigma^2_{RCS'}$], integral time scale [$\tau_{RCS'}$], skewness [$S_{RCS'}$] and kurtosis [$K_{RCS'}$].

Doppler lidar and Elastic lidar analysis: Since you make some effort to quantify the influence of noise on the statistical parameters derived from these two systems, it would be beneficial to *discuss how this impacts your interpretation*, e.g include time-height plots of the correction factor or relative correction, relative importance in determining PBLH, how much temporal averaging is required to obtain good results.

We thank the reviewer 1 for this comment, but as aforementioned in this paper we do not have as objective the PBLH detection for different remote sensing systems. In previous articles (e.g. Bravo-Aranda et al, 2017; Moreira et al., 2018a) we performed a comparison between the PBLH obtained from different remote sensing systems (MWR,Elastic lidar and Doppler lidar), as well as, a discussion about which factors can influence the PBLH generated from data of these systems.

What is the minimum integral time scale that the DL and EL can measure? Is it the acquisition time that allows you to observe turbulence throught the PBL, or is it more likely to be a function of the instrument sensitivities?

The integral time scale (the minimum time where turbulent events are connected) has a minimum acceptable value coincident with the acquisition time of each system (Pal et al., 2010). Thus, in our case the nominal acquisition time is 1 s for the elastic lidar, but we use 2 s for both elastic and Doppler lidar because higher temporal resolutions do not allow for observing turbulence with our instruments. More sensitive systems have lower minimum time, e.g. High Spectral Resolution Lidar utilized by McNicholas et al. (2014).

Case study 2: Did you try cloud-screening EL data before calculating EL parameters? The PBLH from EL would agree much better with PBLH from MWR in Figure 13, and maybe Figure 14 (it is hard to tell with the scales used). Clouds should also be visible in DL data.

We thank the reviewer 1 for this comment. No, any cloud-screening method was not applied before calculating the EL parameters due to the main objective of this manuscript is not to improve the quality of PBLH detection from EL data, but to show how clouds and Saharan dust layers can influence in the PBL characterization when aerosols are used as tracers.

**Technical comments**

Line 36: What do you mean by cyclic processes?

Line 37: Large variability of what?

We thank the Reviewer 1 for these two questions. In order to clarify these points, the text has been changed as follows:

*(Page 1, Line 36)*

*"…is mainly characterized by turbulent processes and a daily evolution cycle…"*

Line 39: Surface heating is unlikely to impact the upper troposphere.

We thank the Reviewer 1 for this comment. In order to clarify this point, the text has been changed as follows:

*(Page 1, Line 40)*

*"This process intensifies the convection and, thus, the ascending warm air masses heat the air masses situated in the upper regions of troposphere, originating the Convective Boundary Layer…"*

Line 84: Distinct?

We thank the Reviewer 1 for this question. In order to clarify this point, the text has been changed as follows:

*(Page 3, Line 85)*

*"…operating at different altitudes…"*

Line 89: Replace 'responsible of ' with 'responsible for'.

Done.

Line 98: Explain '(s and p)'.

Our elastic lidar is polarization-sensitive. In this way, "s" and "p" refers to the parallel channel (p) and the perpendicular one (s). The text has been changed accordingly:

*(Page 3, Line 99)*

*"…532 (parallel and perpendicular polarization)…"*

Line 104: Please include a few more Doppler lidar operating parameters: pulse repetition frequency, telescope focus.

We thank the Reviewer 1 for this comment. In order to clarify this point, the text has been changed as follows:

*(Page 3, Line 106)*

*"It operates at 1.5 µm with pulse energy and repetition rate of 100 µJ and 15 KHz, respectively. This system record the backscattered signal with 300 gates, being the range gate length is 30 m, with the first gate at 60 m. The telescope focus is set to approximately 800 m."*

Line 106: Use 'laser beam pointing at vertical', since the ground surface may not be horizontal!

We thank the Reviewer 1 for this comment. In order to clarify this point, the text has been changed as follows:

*(Page 3, Line 109)*

*"...laser beam is pointed at vertical with respect…"*

Line 108: Replace 'which is part of the MWRNet' with 'which is a member of MWRNet'.

Done.

Line 112: State how many frequencies measured in each band.

We thank the Reviewer 1 for this comment. In order to clarify this point, the text has been changed as follows:

*(Page 4, Line 127)*

*"K-band (water vapor – frequencies: 22.24 GHz, 23.04 GHz, 23.84 GHz, 25.44 GHz, 26.24 GHz, 27.84 GHz, 31.4 GHz) and V-band (oxygen – frequencies: 51.26 GHz, 52.28 GHz, 53.86 GHz, 54.94 GHz, 56.66 GHz, 57.3 GHz, 58.0 GHz)"*

Line 128: Replace 'MWR data analyzes' with 'MWR data analysis'

Done.

Line 130: PBLH not defined yet.

We thank the Reviewer 1 for the comments. In order to clarify this point, the text has been changed as follows:

*(Page 4, Line 131)*

*"...PBL Height ($PBLH_{MWR}$)…"*

Line 188: Do you mean '(Pal et al., 2010)'?

Yes, the manuscript has been modify accordingly:

*(Page 6, Line 192)*

*"...(Pal et al., 2010)…"*

Line 220: Replace 'Under' with 'Below'.

Done.

Lines 220-221: This sentence does not make sense. Do you mean 'Below the PBLH_MWR, correcting for noise does not have a significant impact on the profile, but is more evident above'?

We thank the Reviewer 1 for this question and we apologize for this mistake. In order to clarify this point, the text has been changed as follows:

*(Page 7, Line 239)*

*"The profiles corrected by -2/3 law do not present notable differences in comparison to uncorrected profiles. On the other hand, the profiles corrected by the first lag correction have significant differences below the $PBLH_{MWR}$, mainly the $\sigma_w^2$, ($S_w$, only in the first 50 m), and some slight differences are evident above $PBLH_{MWR}$."*

Line 261: Define Rn (presumably net surface radiation).

Done:

*(Page 8, Line 266)*

*"…net surface radiation ($R_n$)…"*

Line 320: Do you mean '(Ansmann, 2010)'?

Yes, we do and the text has been changed as follows:

*(Page 10, Line 325)*

*"…(Ansmann et al., 2010)…"*

Figure 4: Autocovariance from DL? What are the units for variance and skewness?

Figures 5,7: Profiles from which instrument, and from which location? At what time, and on what day? What height is the surface?

Figure 6: Autocovariance from EL? What are the units for variance, skewness and kurtosis?

We thank the Reviewer 1 for these questions. In order to clarify these points, the figures has been changed as follows:

In figure 4 the title has been changed of "Autocovariance" to "Autocovariance function of $w'$"

[Figure]

Figure 4 – Autocovariance function (ACF) of w' at three different heights

In figure 5 the units of Variance (m/s²) and Skewness (a.u.) have been added, as well as, information about time, location of measurement.

[Figure]

Figure 5 – A - Vertical profile of Integral time scale ($\tau_{w'}$). B - Vertical profile of variance ($\sigma^2_{w'}$). C - Vertical profile of Skewness. ($S_{w'}$)

In figure 6 the title has been changed of "Autocovariance" to "Autocovariance function of $RCS'$"

[Figure]

Figure 6 – Autocovariance of RCS' to three different heights

In figure 7 the units of Variance (m²/s²), Skewness (a.u.) and Kurtosis (a.u.) have been added, as well as, information about time, location of measurement.

[Figure]

Figure 7 – A- Vertical profile of Integral time scale ($\tau_{RCS'}$). B - Vertical profile of variance ($\sigma^2_{RCS'}$). C - Vertical profile of Skewness ($S_{RCS'}$). D - Vertical profile of Kurtosis ($K_{RCS'}$).

Figures 8,11: Which instrument are panels A-C from? Are the black lines (temperature) from the MWR retrieval? Is it more appropriate to plot variance in log scale?

We thank the Reviewer 1 for these questions. In order to clarify these points, the figures has been changed as follows:

In figure 8-A, B and C a label has been added.

[Figure]

Figure 8 – A – integral time scale $[\tau_{w'}]$, B – variance $[\sigma_{w'}^2]$, C – skewness $[S_{w'}]$, D – net radiation $[R_n]$, E – Air surface temperature [blue line] and surface relative humidity $[RH$ – orange line]. In A, B and C black lines and white stars represent air temperature and $PBLH_{MWR}$, respectively.

In figure 12-A, B and C a label has been added.

[Figure]

Figure 12 - A – integral time scale [$\tau_{w'}$], B – variance [$\sigma^2_{w'}$], C – skewness [$S_{w'}$], D – net radiation [$R_n$], E – Air surface temperature [blue line] and surface relative humidity [$RH$ – orange line]. In A, B and C black lines and white stars represent air temperature and $PBLH_{MWR}$, respectively.

Figure 9,12: Which instrument is this figure from?

We thank the Reviewer 1 for this question. As indicated in the title of these figures, the RCS profile is obtained from MULHACEN (the Raman lidar system) data.

This is a time-height plot of RCS, not a profile.

We thank the Reviewer 1 for this comment. In order to clarify this point the label of figures has been changed as follow:

*"Time-Height plot of RCS ..."*

[revised manuscript text omitted]

---

## Author Response (AR2)

**Analyzing the turbulent Planetary Boundary Layer by remote sensing systems: Doppler wind lidar, aerosol elastic lidar and microwave radiometer *by* Gregori de Arruda Moreira et al.**

**Author's response.**

We thank the anonymous reviewers for their comments, corrections and suggestions, which have helped to improve the quality of the manuscript. According to the referees' reports, the following changes have been performed on the original manuscript and a point-by-point response is included below.

**Reviewer 1**

The paper has improved significantly. Thank you for the explanations. I understand now much better how the different measurement techniques are applied together in order to gather combined information for analysis of PBL behavior. But I also see now that I was quite distracted by the technical description and I was actually missing a clear overview about the paper's idea. Here some suggestions:

1) It may be helpful to describe in one or two sentences what you mean by *"PBL behavior"*. This expression is quite general and leaves a lot of room for interpretation, because it is not well defined and can be interpreted in many differently ways. Maybe you can just add a list of parameters and/or boundary layer properties which you want to derive synergistically. From my current understanding of the paper that would be things like heating/cooling source, aerosol source (ground emission or long-range transport), presence of top down mixing... Those are mentioned in the introduction, but a condensed list of the target parameters would facilitate the understanding of the paper's intention and focus dramatically.

We thank the Reviewer 1 for these comments/suggestions. In order to clarify the question raised on terminology we replaced "*PBL behavior*" by "*analysis of the PBL*", because the main idea is to talk about the different processes that occur in the PBL and use them to characterize this layer. In addition, we replaced the term "*synergy*" by "*combination*". We consider this term more appropriate because the results generated from the different instruments are used in a complementary way leaving the exploitation of synergies for a future work.

The table 2 presents a list of all variables analyzed individually and their respective products.

Table 2 – Products and their respective meaning, provided by each system

| Product | System | Meaning |
|---|---|---|
| $\tau_{w'}(z)$ | Doppler lidar | Measurement in time of length of turbulent eddies |
| $\sigma^2_{w'}(z)$ | Doppler lidar | Turbulent Kinetic Energy |
| $S_{w'}(z)$ | Doppler lidar | Direction of turbulent movements |
| $PBLH_{Doppler}$ | Doppler lidar | Top of CBL obtained from variance threshold method |
| $\tau_{RCS'}(z)$ | Elastic lidar | Measurement in time of length of turbulent eddies |
| $\sigma^2_{RCS'}(z)$ | Elastic lidar | Homogeneity of aerosol distribution |
| $S_{RCS'}(z)$ | Elastic lidar | Aerosol motion (S < 0 ➔ Downdrafts, S> 0➔ Updrafts) |
| $K_{RCS'}(z)$ | Elastic lidar | Level of aerosol mixing (K < 3 ➔ Well-Mixed, K > 3 ➔ Low Mixing) |
| $PBLH_{Elastic}$ | Elastic lidar | Top of aerosol layer obtained from variance method |
| $PBLH_{MWR}$ | MWR | Top of CBL/SBL layer obtained from Potential Temperature |

2) Even better would be a new figure that shows which input parameters are needed for which of these boundary layer properties. It is described in the text, but a figure like this would make the "synergy" and the idea of the work more obvious for the reader.

We thank the Reviewer 1 for this comment. In order to clarify this point, the table 2 (presented above) was added in the main document.

**Reviewer 2**

**General comments**

This manuscript presents results from the SLOPE campaign in Granada, Spain, in which the objective was to obtain closure between remote sensing and in-situ measurements. For this manuscript, the focus is on characterizing the planetary boundary layer using a Doppler lidar, multi-wavelength lidar (MULHACEN), and a profiling microwave radiometer, all operating at high temporal resolution (2 seconds). The authors investigate the use of fluctuations in aerosol number density from the elastic system (EL), vertical velocity fluctuations obtained from the Doppler lidar (DL), and potential temperature profiles retrieved from the microwave radiometer (MWR), to identify the boundary layer height (PBLH).

As stated in the first review, some of the methodology is relevant, and the influence of random error introducing extra noise in higher-order moments is explored using suitable techniques, but a major issue was that the manuscript did not have a concrete focus and conclusion, and without these, did not present anything new.

The authors now state that the focus of the paper is 'the synergetic combination of information from different remote sensing systems that are sensitive to different tracers'. This would present something new and useful to the scientific community but there is minimal and insufficient discussion presented in the manuscript in its current state. The comments outlined in the first review have not yet been adequately addressed, and so the manuscript is not yet ready for publication.

We thank the Reviewer 2 for this comment. In order to clarify this point we changed the term "synergy" by "combination", because we believe that this term is more pertinent. Thus, we show that the combination of different remote sensing systems can provide a detailed picture on the *PBL* turbulent features. In addition, we performed a more detailed discussion about the influence of noise in our results, mainly at 532 nm (Elastic lidar). Some part of the text have been completely re-written and additional results are presented just to show that the focus of the paper is the description of the turbulent behavior of the PBL by means of higher order moments applied to the 2 s profiles retrieved from Doppler Lidar and Elastic Lidar, specially showing the feasibility of Elastic Lidar signals at 532 nm.

**Major comments**

The authors state that the focus of the paper is on the synergistic combination of different methods to determine PBLH, but there is still insufficient discussion in the main text. There is little suggestion on how the various retrieved parameters could be combined, or how the relative magnitudes of their uncertainties could influence the combination.

We thank the Reviewer 2 for this comment. In order to clarify this point the term *"synergy"* was replaced by *"combination"*, because the results generated from the different instruments are complementary. We consider this term more appropriate because the results generated from the different instruments are used in a complementary way leaving the exploitation of synergies for a future work. The discussion about the combined variables was rewritten in order to improve the text. (Section 4.2 – lines: 309 – 472)

"The EL and DL parameters are calculated over 1-hour periods. Is this 1-hour timescale suitable during rapidly varying conditions such as during the morning growth of the boundary layer?" This question is asking whether a 1 hour timescale is suitable when, during the morning growth, a particular region may have been calm for 30 minutes, and then strongly turbulent for 30 minutes. If changing the timescales has an impact, then this is important information to include in the manuscript, e.g. how does the noise reduce the integral time scale, and is this SNR-dependent? The abstract states that there is "low influence of noise", so how can both od these statements be correct?

We thank the Reviewer 2 for this comment. The time interval of 1-hour provided us values of integral time scale, considerably higher than 2 s in all situations demonstrated in the main document.
The figure 1 presents the variation of integral time scale to different time intervals. It is possible to observe the influence of interval duration. Thus, as time interval increases, the integral time scale also increases, and only with 1 hour of time interval, all points in integral time scale are at least ten times higher than DL time acquisition. Thus, we keep the study based on 1-hour timescale in order to obtain reliable analyses although in some cases we can miss faster changes of the PBL.

[Figure]

Supplementary Material - Figure 1 – Integral Time Scale obtained from

The influence of noise is shown in figure 5 of the main document. As can be seen there the uncorrected profiles present some underestimation and in this way the application of the corrections, mainly the first lag correction, solve this underestimation.

"What is the impact if you change the averaging period, and why was 1-hour chosen when the MWR data are averaged over 30 minutes?". The authors state that using a different timescale for MWR parameters does not interfere in the analysis, yet the focus is the synergistic combination of the various retrievals? The MWR retrieval is much smoother in time than the lidar retrievals, so what is the likely impact when combining them?

We thank the Reviewer 2 for this comment. The value of $PBLH_{MWR}$ is not combined with high frequency variables.It is only used as indicator of height of $CBL$ and $SBL$ in order to differentiate the sublayers in the $PBL$ region.

The manuscript requires a much more rigorous but short description of the processes driving turbulent mixing in the boundary layer.

We thank the Reviewer 2 for this comment. This section was rewritten in order to improve the discussion. We expect that the new redaction solve the reviewer's criticism.

*(lines 331-339)*
*"The skewness of $w'$ ($S_{w'}$) is shown in Figure 11-C. The $S_{w'}$ is directly associated with the direction of turbulent movements. Thus, positive values (red regions) correspond with a surface-heating-driven boundary layer, while negative (blue regions) ones are associated to cloud-top long-wave radiative cooling. During the stable period, there is predominance of low absolute values of $S_{w'}$. Nevertheless, as air temperature increases (transition from stable to unstable period), $S_{w'}$ values begin to become larger. Air temperature begins to decrease around 18:00 UTC, and there is a reduction of $S_{w'}$, so that, the generation rate of convective turbulence decreases. Therefore, the turbulence cannot be maintained against dissipation, then the CBL becomes a SBL covered by the RL. So, the reduction observed in the $PBLH_{MWR}$ is due to the SBLH detection."*

*(lines 383-391)*
*"The positive values of $S_{RCS'}$ observed in the lowest part of profile and above the $PBLH_{Elastic}$ represents the updrafts aerosol layers. The negative values of $S_{RCS'}$ indicates the region with low aerosol concentration due to clean air coming from free troposphere (FT)."*

The response from the authors does not satisfy this requirement, is far too long, and contains many factual errors. For instance, air temperature is not directly related to RN and Sw'. The abstract states (lines 22-24) "Furthermore, we show how some meteorological variables such as air temperature, aerosol number density, vertical wind speed, relative humidity and net radiation might influence the turbulent PBL dynamic" but the main text does not discuss how any of these parameters influence the PBL, and in any case it is not clear to me how, in most situations, the aerosol number density would influence the turbulent PBL dynamic.

We thank the Reviewer 2 for this comment. Such text was removed from the document. We apologize for this misinterpretation.

**Minor comments**

DL analysis: The time-height plots provided in the supplementary material are not satisfactory. The upper panel displays vertical velocity, not wind speed and appears to have been drastically smoothed or averaged compared to the lower panel, which is not backscatter but potentially attenuated backscatter. From the system configuration information provided by the authors (telescope focus) it is unlikely that the 'attenuated backscatter' field has been corrected for the telescope focus, hence the request to provide the signal (SNR+1) instead. Please plot both the vertical velocity and signal (SNR+1) at the original resolution without averaging. If you plot the SNR, you can then see that all velocity data above about 2 km is likely to be noise.

We thank the Reviewer 2 for this comment. In the figures 2-5, presented below, the background correction was applied (Manninen et al., 2016).

[Figure]

Supplementary Material - Figure 2 – Doppler lidar SNR+1 Intensity

[Figure]

Supplementary Material -Figure 3 – Vertical wind Speed obtained from Doppler lidar

[Figure]

Supplementary Material - Figure 4 – Doppler lidar SNR+1 Intensity

[Figure]

Supplementary Material - Figure 5 - Vertical wind speed obtained from
Doppler lidar data

The criterion used to select the regions where our methodology presents valid results is based on the value of the integral time scale, as described in lines 296-298 of main document:

*"The gray areas represents the region where $\tau_w$, is lower than the acquisition time of DL and, therefore, in this region it is not possible to analyze the turbulent processes."*

Thus, despite the gray regions have values of vertical wind speed (supplementary material figures 3 and 5, respectively), these regions were disregarded in our analysis about turbulence because the values of integral time scale are lower than the acquisition time of the *DL*. This criterion has certain similarities with choosing velocities (supplementary material figures 3 and 5, respectively) with SNR+1 more than a certain threshold (supplementary material figures 2 and 4), as suggested by the referees.

"EL analysis: 'Is it safe to assume the two-way transmittance is negligible?" and "what are the typical molecular, aerosol and total extinction values for the cases shown here?" The two-way transmittance at a wavelength 532 nm is less than 0.98 by 2 km above sea level from molecular extinction alone. The AOD is also > 0.1 for 19th May 2016, and over 0.3 for 8th July 2016 (at a wavelength of 500 nm, from AERONET data). These values are not negligible. They may not be sufficient to impact the results, but whether this is the case should be discussed. Does this attenuation impact the noise characteristics and the integral time scale? The new figures 10 and 14 do not aid the interpretation and are not necessary for the manuscript.

We thank the Reviewer 2 for this comment. In order to clarify this point, we present in the revised version of the manuscript the comparison between the analyses based on the use of the wavelengths 1064 and 532 nm, in the sub-section 4.1. In this discussion we use the analyses based on the wavelength of 1064 nm as a reference, considering the negligible influence of the extinction and the molecular component of the backscatter coefficient (figure 10). The comparison between the autocovariance functions derived for 1064 and 532 nm (figures 6 and 7, respectively) evidences the larger noise impact on the wavelength of 532 nm. This larger noise value is due to the impact of the molecular signal at 532 nm, but also in this case because of the extinction by aerosol up to this height. We have estimated two-way transmittances (accounting for both aerosol and molecules) for the two studied cases, obtaining 0.85 and 0.79 respectively.

This characteristic affects the values of high order moments present in figure 9, however, in the same picture is evidenced that the application of the proposed corrections, mainly first lag correction, can significantly reduce the influence of noise and provide results rather comparable to those obtained using the wavelength of 1064 nm in the turbulence analyses.

We included this discussion in the manuscripts, lines 282 – 288:

*"It is evident the larger contribution of $\beta_{Aerosol}^{1064}$ to the total $\beta$ at 1064 nm in comparion with the behavior at 532 nm, generating the higher values of noise at 532 nm in comparison with 1064 nm. This higher noise values are also due to higher extinction (by both aerosol and molecules) at 532 nm, producing a lower two-way transmittance. As we used Elastic lidar technique, we could not calculate aerosol extinction profiles, but an estimation of these transmittances was done on the basis of Klett method (Klett, 1985). With this method, a constant lidar ratio value was constrained for each profile using the AOD derived from a collocated AERONET Sun-photometer [Guerrero-Rascado et al., 2008]. Using these constrained lidar ratios, the transmittances were calculated together with aerosol backscatter profiles, integrated up to 2.5 km. The estimated two-way transmittance was 0.85 for the case analyzed in this subsection (19th May)."*

Supplementary material, Figs. 5 and 6, do not show time-height plots, only two profiles, so it is still not possible to evaluate whether these parameters provide a reliable guide to the boundary layer development.

We thank the Reviewer 2 for this comment. In order to clarify this point we provided 4 hours (4 figures) of the profiles of high order moments.

From the variance profiles is possible to observe the evolution of top of aerosol layer ($PBLH_{Elastic}$), which is practically coincident with $CBL$ height in the first two hours (Supplementary Material Figure 6 and Main document figure 7), but due to the presence of lofted aerosol layer at 14 UTC, the $PBLH_{Elastic}$ and $CBL$ move away, as can be observed in main document figure 12. From the skewness profiles it is possible to follow the ascension of the entrainment zone (inflection point in skewness profile), as well as, the regions dominated by downdrafts and updrafts, which influence directly in the profiles of next hour. The kurtosis profiles show the variation of level of mixing, so that the region with low level of mixing due to entrainment of air from $FT$ follows the ascension of $PBL$, as expected.

[Figure]

Supplementary Material - Figure 6 - Statistical moments obtained from elastic lidar data at 12 to 13 UTC - 19 May 2016. From left to right: variance [$\sigma^2_{RCS'}$], integral time scale [$\tau_{RCS'}$], skewness [$S_{RCS'}$] and kurtosis [$K_{RCS'}$].

[Figure]

Figure 13 – Statistical moments obtained from elastic lidar data (Mulhacen) in Granada at 13 to 14 UTC - 19 May 2016. From left to right: variance [$\sigma^2_{RCS'}$], integral time scale [$\tau_{RCS'}$], skewness [$S_{RCS'}$] and kurtosis [$K_{RCS'}$].

[Figure]

Supplementary Material - Figure 7 - Statistical moments obtained from elastic lidar data at 14 to 15 UTC - 19 May 2016. From left to right: variance [$\sigma^2_{RCS'}$], integral time scale [$\tau_{RCS'}$], skewness [$S_{RCS'}$] and kurtosis [$K_{RCS'}$].

[Figure]

Supplementary Material - Figure 8 - Statistical moments obtained from elastic lidar data at 15 to 16 UTC - 19 May 2016. From left to right: variance [$\sigma^2_{RCS'}$], integral time scale [$\tau_{RCS'}$], skewness [$S_{RCS'}$] and kurtosis [$K_{RCS'}$].

Doppler lidar and Elastic lidar analysis: Since you make some effort to quantify the influence of noise on the statistical parameters derived from these two systems, it would be beneficial to discuss how this impacts your interpretation, e.g. include time-height plots of the correction factor or relative correction, relative importance in determining PBLH, how much temporal averaging is required to obtain good results. You state that your objective is 'to approach a synergetic combination', hence discussing how the influence of noise impacts your interpretation is vital, otherwise there is nothing new presented in this manuscript.

We thank the Reviewer 2 for this comment. In order to clarify this point the sub-section 4.1 have changed as follow:

*"**4.1 Error Analysis**

[revised manuscript text omitted]

*Therefore, in spite of the larger attenuation expected at 532 nm wavelength due to relevant percentage of $\beta_{Molecular}$ in its composition, which generates noisier profiles in comparison with that ones generated from the reference wavelength, the application of the proposed corrections, mainly the first lag, reduce significantly the influence of noise and enable the observation of the same phenomena detected in the high-order moments obtained from 1064 nm. Consequently, the wavelength 532 nm will be applied in the analysis presented in section 4.2. Due to the first lag correction generates a higher impact on the without correction profiles, we adopted such correction in order to be more careful in the analyses of high-order moments obtained from DL and EL data."*

Case study 2: Did you try cloud-screening EL data before calculating EL parameters? The PBLH from EL would agree much better with PBLH from MWR in Figure 13, and maybe Figure 14 (it is hard to tell with the scales used). Clouds should also be visible in DL data. The authors response is "No, any cloud-screening method was not applied before calculating the EL parameters". What happens if you do attempt a simple cloud screening procedure. This is simple to apply and would presumably be used in any synergetic combination?

We thank the Reviewer 2 for this comment. The problem with PBLH detection by EL in case two is not only because of the clouds, but mainly due to the presence of a Saharan dust layer. As we observe in the manuscript, the usual algorithms for PBLH detection do not work well for these cases. It is shown, for example by Bravo-Aranda et al. (2017), that more sophisticated methods using depolarization information may be able to improve this detection. However, it was not the scope of the present work, since we wanted to show the potential of usual 522 nm EL in the different networks (although they have no depolarization channels).
The cloud-screening algorithms may also give not correct results in this case, where the detected cloud might actually be a more intense dust layer. As suggested by referee, we tried with a simple cloud screening procedure, but it marked as clouds some regions that might not be. We checked this information with AERONET cloud screening (as independent validation) at the same time intervals, and the comparison was confusing.
This analysis reinforces the main idea of this paper, that the combination of different instruments and methods is needed for the study of such complex cases. Moreover, it is shown that with the PBLH detection of the different systems we can separate different layers (e.g. the dust layer from the actual CBL).

If the DL telescope focus is set to 800 m then what method do you use to obtain attenuated backscatter profiles from the (SNR + 1) profile? Therefore, in the supplementary material it would be more appropriate to present the time-height plots of vertical velocity and (SNR + 1), which was what was originally requested.

We thank the Reviewer 2 for this comment. We have not yet implement any focus correction for the DL, as this is not relevant for the retrieved velocities although it is for the attenuated backscatter. Thus, as suggested, intensity (SNR + 1) profiles were included instead of attenuated backscatter.

**Line 32: How do the variables interfere in the process?**

Lines 40-42: Please check and reformulate these sentences. Surface heating is still unlikely to directly impact the upper troposphere. The convective boundary layer does not reach the upper troposphere.

We thank the Reviewer 2 for this comment. Such text was removed from the document. We apologize for this misinterpretation.

Lines 226-229: The methodologies are not used synergistically, even though this is the focus of the paper, and it is not shown or discussed how each variable 'influences' the turbulent PBL behaviour.

We thank the Reviewer 2 for this comment. In order to clarify this point, as mentioned above, the term "synergy" was changed to "combination", because we believe that the results of different instruments are complementary. From the combination of the results we retrieved a detailed picture on the $PBL$ turbulent features.

Many of the figures still have very short captions without enough information. Please include the instrument name, date and the location in the caption.

We thank the Reviewer 2 for this comment. In order to clarify this point, all figures mentioned below have its captions changed.

Figure 4: Is this autocovariance from DL?

We thank the Reviewer 2 for this comment. In order to clarify this point the caption of this picture was changed as follow:

"Autocovariance function (ACF) of $w'$, obtained from Doppler lidar at three different heights on 19th May 2016 at 08-09 UTC in Granada."

Figures 5,7: Profiles from which instrument, and from which location? At what time, and on what day? What height is the surface? Please include this information in the caption

Figures 5, 7-10,12-14: The captions do not state which instrument the data comes from. Please include the instrument names and the location in the caption. Where applicable, state which data comes from which instrument.

Figures 11, 15, 16: The caption states 'elastic lidar data'. Please include the instrument name and the location in the caption.

We thank the Reviewer 2 for these comments. In order to clarify this point the captions of these figures, that now have new numbers, and that have been changed as follows:

Figure 6 – Autocovariance of $RCS'_{1064}$ obtained from Mulhacen elastic lidar data to three different heights on 19th May 2016 at 12-13 UTC in Granada.

Figure 10 – A- Vertical profile of Integral time scale ($\tau_{RCS'}$). B - Vertical profile of variance ($\sigma^2_{RCS'}$). C - Vertical profile of Skewness ($S_{RCS'}$). D - Vertical profile of Kurtosis ($K_{RCS'}$). All profiles were obtained from Mulhacen elastic lidar data on 19th May2016 in Granada between 12-13 UTC.

Figure 11 – A – integral time scale obtained from Doppler lidar data [$\tau_{w'}$], B – variance obtained from Doppler lidar data [$\sigma^2_{w'}$], C – skewness obtained from Doppler lidar data [$S_{w'}$], D – net radiation obtained from pyranometer data [$R_n$], E – Air surface temperature [blue line] and surface relative humidity [$RH$ - orange line] both were obtained from surface sensors. All profiles were acquired on 19th May 2016 in Granada. In A, B and C black lines and white stars represent air temperature and $PBLH_{MWR}$, respectively.

Figure 12 – Time-Height plot of RCS obtained on 19 May 2016 in Granada. Pink stars represent the $PBLH_{MWR}$, black stars represent the $PBLH_{Elastic}$ and blues stars represent the $PBLH_{Doppler}$.

Figure 13 – Statistical moments obtained from 532 nm wavelength data of elastic lidar (Mulhacen) in Granada at 13 to 14 UTC - 19 May 2016. From left to right: variance [$\sigma^2_{RCS'}$], integral time scale [$\tau_{RCS'}$], skewness [$S_{RCS'}$] and kurtosis [$K_{RCS'}$].

Figure 15 - A – integral time scale from Doppler lidar data [$\tau_{w'}$], B – variance from Doppler lidar data [$\sigma^2_{w'}$], C – skewness from Doppler lidar data [$S_{w'}$], D – net radiation from pyranometer data [$R_n$], E – Air surface temperature [blue line] and surface relative humidity [$RH$ – orange line] from surface sensor data. All profiles were obtained in Granada on 08 July 2016. In A, B and C black lines and white stars represent air temperature and $PBLH_{MWR}$, respectively.

Figure 16 – Time-Height plot of RCS obtained from Mulhacen elastic lidar data on 08 July 2016 in Granada. Pink stars represent the $PBLH_{MWR}$, black stars represent the $PBLH_{Elastic}$ and blues stars represent the $PBLH_{Doppler}$.

Figure 17 - Statistical moments obtained from 532 nm wavelength data of elastic lidar (Mulhacen) in Granada between 11-12 UTC on 08th July 2016. From left to right: variance [$\sigma^2_{RCS'}$], integral time scale [$\tau_{RCS'}$], skewness [$S_{RCS'}$] and kurtosis [$K_{RCS'}$].

Figure 19 - Statistical moments obtained from 532 nm wavelength data of elastic lidar (Mulhacen) in Granada between 12 -13 UTC on 08 July 2016. From left to right: variance [$\sigma^2_{RCS'}$], integral time scale [$\tau_{RCS'}$], skewness [$S_{RCS'}$] and kurtosis [$K_{RCS'}$].

[revised manuscript text omitted]
\prime}$). B - Vertical profile of variance ($\sigma^2_{RCS\prime}$). C - Vertical profile of Skewness ($S_{RCS\prime}$). D - Vertical profile of Kurtosis ($K_{RCS\prime}$). All profiles were obtained from Mulhacen elastic lidar data on 19th May2016 in Granada between 12-13 UTC.

[Figure]

Figure 10 – A- Vertical profile of Integral time scale ($\tau_{RCS\prime}$). B - Vertical profile of variance ($\sigma^2_{RCS\prime}$). C - Vertical profile of Skewness ($S_{RCS\prime}$). D - Vertical profile of Kurtosis ($K_{RCS\prime}$). All profiles were obtained from Mulhacen elastic lidar data on 19th May2016 in Granada between 12-13 UTC.

[Figure]

Figure 11 – A – integral time scale obtained from Doppler lidar data [$\tau_{w'}$], B – variance obtained from Doppler lidar data [$\sigma^2_{w'}$], C – skewness obtained from Doppler lidar data [$S_{w'}$], D – net radiation obtained from pyranometer data [$R_n$], E – Air surface temperature [blue line] and surface relative humidity [$RH$ - orange line] both were obtained from surface sensors. All profiles were acquired on 19th May 2016 in Granada. In A, B and C black lines and white stars represent air temperature and $PBLH_{MWR}$, respectively.

[Figure]

Figure 12 – Time-Height plot of RCS obtained on 19 May 2016 in Granada. Pink stars represent the $PBLH_{MWR}$, black stars represent the $PBLH_{Elastic}$ and blues stars represent the $PBLH_{Doppler}$.

[Figure]

Figure 13 – Statistical moments obtained from 532 nm wavelength data of elastic lidar (Mulhacen) in Granada at 13 to 14 UTC - 19 May 2016. From left to right: variance [$\sigma^2_{RCS'}$], integral time scale [$\tau_{RCS'}$], skewness [$S_{RCS'}$] and kurtosis [$K_{RCS'}$].

[Figure]

Figure 14 – Statistical moments obtained from 1064 nm wavelength data of elastic lidar(Mulhacen) in Granada at 13 to 14 UTC - 19 May 2016. From left to right: variance [$\sigma^2_{RCS\prime}$], integral time scale [$\tau_{RCS\prime}$], skewness [$S_{RCS\prime}$] and kurtosis [$K_{RCS\prime}$].

[Figure]

Figure 15 - A – integral time scale from Doppler lidar data [$\tau_{w'}$], B – variance from Doppler lidar data [$\sigma_{w'}^2$], C – skewness from Doppler lidar data [$S_{w'}$], D – net radiation from pyranometer data [$R_n$], E – Air surface temperature [blue line] and surface relative humidity [$RH$ – orange line] from surface sensor data. All profiles were obtained in Granada on 08 July 2016. In A, B and C black lines and white stars represent air temperature and $PBLH_{MWR}$, respectively.

[Figure]

Figure 16 – Time-Height plot of RCS obtained from Mulhacen elastic lidar data on 08 July 2016 in Granada. Pink stars represent the $PBLH_{MWR}$, black stars represent the $PBLH_{Elastic}$ and blues stars represent the $PBLH_{Doppler}$.

[Figure]

Figure 17 - Statistical moments obtained from 532 nm wavelength data of elastic lidar(Mulhacen) in Granada between 11-12 UTC on 08th July 2016. From left to right: variance $[\sigma^2_{RCS'}]$, integral time scale $[\tau_{RCS'}]$, skewness $[S_{RCS'}]$ and kurtosis $[K_{RCS'}]$.

[Figure]

Figure 18 - Statistical moments obtained from 1064 nm wavelength data of elastic lidar(Mulhacen) in Granada between 11-12 UTC on 08th July 2016. From left to right: variance $[\sigma^2_{RCS'}]$, integral time scale $[\tau_{RCS'}]$, skewness $[S_{RCS'}]$ and kurtosis $[K_{RCS'}]$.

[Figure]

Figure 19 - Statistical moments obtained from 532 nm wavelength data of elastic lidar (Mulhacen) in Granada between 12 -13 UTC on 08 July 2016. From left to right: variance [$\sigma^2_{RCS'}$], integral time scale [$\tau_{RCS'}$], skewness [$S_{RCS'}$] and kurtosis [$K_{RCS'}$].

[Figure]

Figure 20 - Statistical moments obtained from 1064 nm wavelength data of elastic lidar (Mulhacen) in Granada between 12 -13 UTC on 08 July 2016. From left to right: variance [$\sigma^2_{RCS'}$], integral time scale [$\tau_{RCS'}$], skewness [$S_{RCS'}$] and kurtosis [$K_{RCS'}$].

---

## Author Response (AR3)

**Analyzing the turbulent Planetary Boundary Layer by remote sensing systems: Doppler wind lidar, aerosol elastic lidar and microwave radiometer *by* Gregori de Arruda Moreira et al.**

**Author's response.**

We thank the anonymous reviewers for their comments, corrections and suggestions, which have helped to improve the quality of the manuscript. According to the referees' reports, the following changes have been done in the original manuscript and a point-by-point response is included below.

In order to show that this paper is a nice contribution to demonstrate the broad spectrum of the EARLINET special issue the following phrases have been added:

(Lines 96 – 101)

*"One of the goals is to show the feasibility of using EL at 532 nm, considering the widespread use of lidar systems based on laser emission at this wavelength in different coordinated networks, like as EARLINET (Pappalardo et al., 2014) and LALINET – Latin American LIdar Network (Guerrero-Rascado et al., 2016). In addition, this study shows the variety of application that can be done with EARLINET data applying some simple changes in the data acquisition procedures."*

**Reviewer 1**

**General comments**

This manuscript presents results from the SLOPE campaign in Granada, Spain, in which the objective was to obtain closure between remote sensing and in-situ measurements. For this manuscript, the focus is on characterizing the planetary boundary layer using a Doppler lidar, multi-wavelength lidar (MULHACEN), and a profiling microwave radiometer, all operating at high temporal resolution (2 seconds). The authors investigate the use of fluctuations in aerosol number density from the elastic system (EL), vertical velocity fluctuations obtained from the Doppler lidar (DL), and potential temperature profiles retrieved from the microwave radiometer (MWR), to identify the boundary layer height (PBLH). As stated in the first and second review, the methodology is relevant, and the influence of random error introducing extra noise in higher-order moments has merit and is explored using suitable techniques.

The manuscript has now been improved significantly, with a clearer focus, and explores the impact of applying the elastic lidar methodology at different wavelengths. New Figures 9 and 10 now show that, although backscattering coefficients are wavelength-dependent (molecular and aerosol), the methodology and correction procedure can account for these differences. It is now clear that the higher moments exhibit more correction at 532 nm but the methodology used to derive PBLH from elastic lidar is not unduly sensitive to the wavelength used, at least.

The new supplementary figures are much better, and clearly display where reliable data may be obtained. However, there are still a few issues for the authors to address before the manuscript is suitable for publication.

This question was asked previously: "The EL and DL parameters are calculated over 1-hour periods. Is this 1-hour timescale suitable during rapidly varying conditions such as during the morning growth of the boundary layer?" This question is asking whether a 1 hour timescale is suitable when, during the morning growth, a particular region may have been calm for 30 minutes, and then strongly turbulent for 30 minutes. What happens to the integral timescale for both EL and DL properties when you include atmospheric regions with very different turbulent attributes (e.g calm and convective) within the same averaging period? With one hour averaging, you will always miss the rapid growth of the CBL. I understand you want to capture the integral time scale, but how can you do this when the turbulence characteristics themselves are changing?

We thank the Reviewer for this comment. Answer Figure 1-A (case study I of the main document) shows a situation with a fast growth of the PBL during 30 min followed by 30 min interval with a PBLH almost constant. In figures 1-B and 1-C we show the analyses of this case split in two 30 min intervals corresponding to the two different situations, fast growth and almost constant PBLH. In both cases the integral time scale is lower than that computed for the whole 1 hour interval and the profiles of Skewness and Kurtosis are rather noisy, thus complicating the observation of determined phenomena, although the profiles are very similar. So it is evident that the analyses of intervals bellow 1 hour are not good enough, the degradation of the analyses increases with the reduction of the considered interval as can be seen in the figures (15 min [Fig. 1-D], 10 min [Fig. 1-E] and 5 min [Fig. 1-F]). In this sense, it is evident that the features of our equipment does not allow the detection, with appropriate quality, of turbulent events with a temporal resolution lower than 1 hour. So when the PBL present faster changes we can only observe the average behavior of the turbulence with this time window.

[Figure]

Answer Figure 1 – Statistical moments obtained from 532 nm wavelength data of elastic lidar (Mulhacen) in Granada - 19 May 2016.

**Minor comments**

Lines 42-47. Please check and reformulate these sentences.

We thank the Reviewer for this comment. In order to clarify this point the text has been changed as follow:

(Lines 42 - 48)

*"In an ideal situation, some instants after sunrise, the ground surface temperature increases due to the positive net radiative flux ($R_n$). This process intensifies the convection, where there is an ascension of warm air masses, causing the downward displacement of colder air masses and consequently originating the Convective Boundary Layer (CBL) or Mixing Layer (ML). Such layer has this name due to the mixing process generated by the ascending air parcels. Slightly before sunset, the gradual reduction of incoming solar irradiance at the Earth's surface causes the decrease of the positive $R_n$ and, consequently, its sign change. In this situation, there is a reduction of the convective processes and a weakening of the turbulence."*

Line 78: Replace 'turbulenc' with 'turbulence'.

 Done

Line 86: The convective PBL is the CBL.

We thank the Reviewer for this comment. In order to clarify this point the text has been changed as follow:

(Line 87)

*"… PBL height (PBLH) during the convective period …"*

Line 96: Replace 'realibility' with 'reliability'. Explain why measurements at 532 nm should be more reliable, or suggest removing this sentence.

We thank the Reviewer for this comment. In order to clarify this point the text has been changed as follow:

(Lines 97 - 99)

*"… considering the widespread use of lidar systems based on laser emission at this wavelength in different coordinated networks, like as EARLINET (Pappalardo et al., 2014) and LALINET – Latin American LIdar Network (Guerrero-Rascado et al., 2016)."*

Lines 129-130: Suggest replacing 'This system record the backscattered signal with 300 gates, being the range gate length 30 m, with the first gate at 60 m' with 'This system records the backscattered signal with a range resolution of 30 m in 300 range gates with the first range gate starting at 60 m from the instrument.'

Done

Line 131: Suggest stating 'The instrument was operated in vertical stare mode with a temporal resolution of 2 s'. Using the phrase 'with respect to the ground surface' could mean that, on a sloping surface, you imply you are pointing normal (90 degrees) to the surface. Pointing vertically is unambiguous and doesn't require the qualifier.

We thank the Reviewer for this comment. In order to clarify this point the suggested change has been applied.

Line 168: Replace 'performed campaign of comparison' with 'intercomparison campaign'.

Done

Line 173: Replace 'allow to estimate the CBL height' with 'allows the estimation of the CBL height'.

Done

Lines 179, 182, 186: σw2, σRCS, q', are used before being defined.

We thanks the Reviewer for this comment. In order to clarify this point the text has been changed as follow:

(Line 184)

"… the variance of vertical wind speed ($\sigma_w^2$) …"

(Lines 187-188)

"…the variance of Range Corrected Signal ($\sigma_{RCS}^2$) …"

(Lines 191)

"…gathered data $[q(z,t)]$ with a temporal resolution …"

Line 179: Do you mean the PBLH_Doppler is attributed to the height where σw2 drops below a pre-determined threshold?

Yes, exactly. In order to clarify this point the text has been changed as follow:

(line 184)

"… ($\sigma_w^2$) is lower than a determinate threshold …"

Line 191: Replace 'isthe' with 'is the'.

Done

Line 222: Replace 'depends' with 'depend'.

Done

Lines 242-244: Suggest rewriting this phrase as it is unclear. You could use ' in this study we evaluate using RCS532 fluctuations to determine turbulence following the procedure described in Figure 3. This EL methodology is very similar to that described earlier for DL.'

Done

Line 257 and elsewhere: Suggest using 'below the PBLH' rather than 'under the PBLH'.

Done in line 264 and line 270

Line 259: Replace 'relation the other' with 'relation to the other'.

Done

Lines 263-265. This statement is only true in high SNR conditions - figure 5 only shows data within 1200 m of the instrument.

We thank the Reviewer for this comment. In order to clarify this point the text has been changed as follow:

(Lines 270-271)

"…Therefore, considering high Signal-to-Noise Ratio (SNR) conditions, although the presence of ε…"

Lines 272-273: Suggest replacing 'its spread use in observation network with higher reliability than 1064 nm' with 'its widespread use in observation networks'.

We thank the Reviewer for this comment. In order to clarify this point the suggested change has been applied.

Lines 274-275: Suggest replacing 'As expected, in both cases the increase of height produces the increase of ε,' with 'As expected, ε increases with range'.

Done

Lines 278-283: The difference in noise levels between the two wavelengths depends on the SNR at each wavelength, which is more likely to be determined by the laser output power, filters, and detectors used, at the two wavelengths. Higher molecular extinction at 532 nm can then reduce SNR relative to the 1064 nm wavelength, as does separation of the molecular and aerosol backscattering. Figure 8 is not necessary for the manuscript.

We thank the Reviewer for this comment. In order to clarify this point the figure 8 has been removed and the text has been changed as follow:

(Lines 286-292)

*"Although the level of influence of $\varepsilon$ in each wavelength depends on the $SNR$ of them (which is associated to technical factors such as laser output power, filters, type of detectors), considering the proposed methodology, to evaluate the composition of each wavelength is also important. The large contribution of $\beta_{Molecular}^{532}$ to the total $\beta$ at 532 nm in comparison with the behavior at 1064 nm, can influence the results obtained from such wavelength, because our methodology is based on the use of $\beta'_{Aerosol}$. In addition, the larger extinction (due to both aerosol particles and molecules) at 532 nm produces a lower two-way transmittance, resulting in the reduction of the $SNR$ values at this wavelength."*

Line 299: SNR is reduced, the noise doesn't increase.

We thank the Reviewer for this comment. In order to clarify this point the figure 8 has been removed and the text has been changed as follow:

(Lines 308-310)

*"…which reduces the $SNR$ of the profiles in comparison with 1064 nm, the application of the proposed corrections, mainly the first lag, reduces significantly such influence and…"*

Lines 303-304: Suggest rewriting this phrase as it is unclear.

We thank the Reviewer for this comment. In order to clarify this point the text has been changed as follow:

(Lines 312-313)

*"The first lag correction was adopted as default because it provides better results than the -2/3 law correction."*

Lines 315-318: It is not clear that the integral time scale can be retrieved throughout the whole PBL, especially if PBLH_MWR is taken as a reference. It is not necessarily true that the grey areas are where T_w' is lower than the DL acquisition time, just that the DL sensitivity is not high enough to measure T_w'. This can be seen in the supplementary material, where SNR is low above 1500 m and the upper portion of the CBL may not be captured during daytime. I would expect T_w' to be just as large here.

We thank the Reviewer for this comment. In order to clarify this point the text has been changed as follow:

(Lines 322-325)

*"Figure 10 (A) shows the integral time scale obtained from $DL$ data ($\tau_{w'}$). The gray area represents the region where it is not possible to analyze the turbulent process using our $DL$ data, either because of the low $SNR$ values, which results in null values of the $\tau_{w'}$, or because the no null $\tau_{w'}$ is smaller than the acquisition time of the $DL$. However, the gray area is located almost entirely above the $PBLH_{MWR}$ (white stars)."*

Lines 319-320: This sentence can be removed.

Done

Lines 327-328: I suggest removing this sentence.

Done

Lines 329-320: Not correct, skewness describes the distribution of the turbulent velocities - positive skewness implies strong but narrow updrafts surrounded by weaker but more widespread downdrafts, and vice versa for negative skewness.

We thank the Reviewer for this comment. In order to clarify this point the text has been changed as follow:

Lines (333-335)

*"The skewness of $w'$ ($S_{w'}$) is shown in Figure 11-C. The $S_{w'}$, describes the distribution of the turbulent velocities. Thus positive $S_{w'}$ implies strong but narrow updrafts surrounded by weaker but more widespread downdrafts, and vice versa for negative $S_{w'}$."*

Line 336: Do you mean that that the MWR method is now selecting for SBLH?

We thank the Reviewer for this comment. In order to clarify this point the text has been changed as follow:

(Lines 341-342)

*"… Thus, the reduction observed in the $PBLH_{MWR}$ is due to the detection of SBL height."*

Lines 344-353: Suggest removing these paragraphs. Some of these statements could replace phrases in lines 329-337.

We thank the Reviewer for this comment. In order to clarify this point the text has been changed as follow:

(Lines 349-351)

*"Figure 10-E presents the values of surface air temperature and surface relative humidity (RH). Air surface temperature has a daily pattern similar to that of $R_n$ and $S_{w'}$. On the other hand, RH is inversely correlated with the temperature."*

Lines 365-367: The PBLH values in Figure 13 don't agree with Figure 12, where there is a large difference between PBLH_MWR and PBLH_Elastic.

We thank the Reviewer for this comment. In order to clarify this point the figures 12 and 13 have been remade as shwon below:

[Figure]

Figure 12 – Statistical moments obtained from 532 nm wavelength data of elastic lidar (Mulhacen) in Granada at 13 to 14 UTC - 19 May 2016. From left to right: variance [$\sigma^2_{RCS'}$], integral time scale [$\tau_{RCS'}$], skewness [$S_{RCS'}$] and kurtosis [$K_{RCS'}$].

Figure 13 – Statistical moments obtained from 1064 nm wavelength data of elastic lidar(Mulhacen) in Granada at 13 to 14 UTC - 19 May 2016. From left to right: variance [$\sigma^2_{RCS\prime}$], integral time scale [$\tau_{RCS\prime}$], skewness [$S_{RCS\prime}$] and kurtosis [$K_{RCS\prime}$].

The text has been changed as follow:

(Lines 366-374)

*"Due to presence of a decoupled aerosol layer at 13:30, the average values of $PBLH_{Elastic}$ and $PBLH_{MWR}$ have a difference of around 500 m. The $\sigma^2_{RCS\prime}$ has small and practically constant values between 1000 and 1400m, evidencing the homogeneity of the aerosol distribution in this region. Starting at 1400 m the value of $\sigma^2_{RCS\prime}$ begins to increase, reaching a positive peak at $PBLH_{MWR}$ , which represents the Entrainment Zone (region characterized by an intense mixing between air parcels coming from CBL and Free Troposphere (FT), causing a high variation in aerosol concentration). The $PBLH_{Elastic}$ observed at approximately 2900 m demonstrate an inherent difficulty of the variance method to detect the PBLH in the presence of several aerosol layers (Kovalev and Eichinger, 2004). Above $PBLH_{Elastic}$ the values of $\sigma^2_{RCS\prime}$ decrease slowly due to location of the lofted aerosol around 2500 m."*

Line 372: Define FT here.

Done

Line 392: Do you refer to the correct figure here?

We thank the Reviewer for this comment. In order to clarify this mistake the text has been changed as follow:

(Lines 392 -393)

*"The results provided by $DL$, pyranometer and $MWR$ data agree with the results observed in figures 12 and 13."*

Line 399. Suggest removing the second phrase of this sentence.

Done

Line 400: Not true according to the figure.

We thank the Reviewer for this comment. In order to clarify this point the text has been changed as follow:

(Lines 399-400)

*"…the greatest part of grey area is situated above the $PBLH_{MWR}$ …"*

Lines 411-414: No clouds are observed in the DL data until 1400 UTC, and it is difficult to prove that the negative skewness extends to 3 km in altitude. Are you sure that all of the white regions are cloud before 1400? One at 1230 and one at 1330 maybe.

We thank the Reviewer for this comment. During this period there is the presence of both middle altitude clouds and very intense dust layers. In order to clarify this point the text has been changed as follow:

(Lines 411 - 415)

*"From Figure 16 we can observe the presence of both middle altitude clouds and very intense dust layers from 12:00 to 15:00 UTC. Such combination contributes to the intense negative values of $S_w$, observed in this period until around 2 km, because, as mentioned previously, $S_w$, is directly associated with the direction of the turbulent movements. The present situation can be considered representative of cloud-top long-wave radiative cooling in the CBL (Ansmann et al., 2010)."*

Lines 415-420. As above, it is clear that the Saharan dust layer is having an impact, but Rn alone is probably not sufficient to attribute negative skewness directly to clouds.

We thank the Reviewer for this comment. In order to clarify this point the text has been changed as follow:

(Lines 417-420)

*"The observation of $S_w$, and $R_n$ between 12:00 and 14:00, as well as, the presence of clouds and geometrically thick dust layers during this same period, reinforces the hypothesis that we have a situation of cloud-top long-wave radiative cooling in the CBL."*

Line 423: See above comment. Not all of the high RCS values (white regions) before 1300 can be attributed to clouds, and there is very little attenuation seen in the profile before 1230.

We thank the Reviewer for this comment. In order to clarify this point the text has been changed as follow:

(Line 425 - 426)

*"…the presence of both middle altitude clouds and very intense dust layers …"*

Line 463: This is only true at high SNR.

We thank the Reviewer for this comment. In order to clarify this point the text has been changed as follow:

(Lines 467-468)

*"…low influence of the noise in high SNR conditions."*

Line 467: Replace 'lower two ways' with 'reduced two-way'.

Done

Figure 12: Replace 'blues stars' with 'blue stars'.

Done

[revised manuscript text omitted]